# Exploring Tradeoffs through Mode Connectivity for Multi-Task Learning

**Zhipeng Zhou**[1], **Ziqiao Meng**[2], **Pengcheng Wu**[1], **Peilin Zhao**[3], **Chunyan Miao**[1*]

[1]Nanyang Technological University
[2]National University of Singapore
[3]School of Artificial Intelligence, Shanghai Jiao Tong University
zzpustcml@gmail.com, zq-meng@nus.edu.sg, peilinzhao@sjtu.edu.cn,
{pengchengwu, ascymiao}@ntu.edu.sg

## Abstract

Nowadays deep models are required to be versatile due to the increasing realistic needs. Multi-task learning (MTL) offers an efficient way for this purpose to learn multiple tasks simultaneously with a single model. However, prior MTL solutions often focus on resolving conflicts and imbalances during optimization, which may not outperform simple linear scalarization strategies [Xin et al., 2022]. Instead of altering the optimization trajectory, this paper leverages mode connectivity to efficiently approach the Pareto front and identify the desired trade-off point. Unlike Pareto Front Learning (PFL), which aims to align with the entire Pareto front, we focus on effectively and efficiently exploring optimal trade-offs. However, three challenges persist: (1) the low-loss path can neither fully traverse trade-offs nor align with user preference due to its randomness, (2) commonly adopted Bézier curves in mode connectivity are ill-suited to navigating the complex loss landscapes of deep models, and (3) poor scalability to large-scale task scenarios. To address these challenges, we adopt non-uniform rational B-Splines (NURBS) to model mode connectivity, allowing for more flexible and precise curve optimization. Additionally, we introduce an order-aware objective to explore task loss trade-offs and employ a task grouping strategy to enhance scalability under massive task scenarios. Extensive experiments on key MTL datasets demonstrate that our proposed method, `EXTRA` (EXplore TRAde-offs), effectively identifies the desired point on the Pareto front and achieves state-of-the-art performance. `EXTRA` is also validated as a plug-and-play solution for mainstream MTL approaches. Code is avaliable at `https://github.com/zzpustc/EXTRA`.

## 1 Introduction

Deep models are widely deployed across various scenarios, with an increasing demand for efficiency, such as reducing computational and storage costs. This growing need necessitates that deep models be versatile. In this context, multi-task learning (MTL) has been extensively explored [Zhang and Yang, 2018, Wang et al., 2013, Zhang et al., 2018]. MTL aims to achieve strong performance across multiple tasks using a single model, in contrast to traditional deep learning approaches where the number of parameters or models often scales with the number of tasks.

In recent years, significant attention has been devoted to optimization-based MTL [Sener and Koltun, 2018, Liu et al., 2021a, Zhou et al., 2025a], which assumes a fixed model architecture comprising a task-shared backbone and multiple task-specific branches. The primary goal in this setting is to

---

[*]Corresponding author.

39th Conference on Neural Information Processing Systems (NeurIPS 2025).

develop optimization algorithms that enable the backbone to effectively extract task-shared features. However, this approach faces two major challenges: conflict and imbalance issues, wherein task gradients may exhibit conflicts or significantly imbalanced norms.

To address these challenges, various solutions have been proposed, including gradient orthogonal projection [Yu et al., 2020], Nash negotiation [Navon et al., 2022], and fair resource allocation mechanisms [Ban and Ji, 2024], etc. Unfortunately, recent studies [Xin et al., 2022, Mueller et al., 2024, Zhou et al., 2025b] have argued that altering the optimization trajectory through combinations of task gradients may be less effective than anticipated. Notably, when dedicated tuning is applied across all methods, these advanced approaches often fail to outperform a simple linear scalarization strategy. These findings suggest the need to explore alternative paradigms for optimization in MTL.

Another research line is Pareto Front Learning (PFL), which aims to directly explore the entire Pareto front, enabling the attainment of Pareto optimality based on user preferences. For instance, inspired by mode connectivity, PaMaL [Dimitriadis et al., 2023] attempts to approach the Pareto front through ensembles of single tasks. Several efficient variants [Tang et al., 2024, Dimitriadis et al., 2024, Chen and Kwok, 2024] have since been proposed, incorporating techniques such as low-rank approximations or mixtures of experts. However, challenges in manifold learning and the scalability of PFL as the number of tasks increases remain unresolved and inapplicable, as we will empirically verify in Section 4.

In this paper, rather than leveraging mode connectivity [Garipov et al., 2018] to achieve PFL, we utilize it to address MTL. Unlike PFL, which optimizes endpoints and attempts to align with the Pareto front in the manifold space, our approach optimizes the curve connecting the endpoints. This requires only two endpoints and a few control points, ensuring that memory and storage costs do not scale with the number of tasks. However, three challenges remain: (1) The low-loss path identified by mode connectivity for MTL is random, lacking both trade-off traversal and alignment with user preferences. (2) Commonly adopted curves (e.g., Bézier curves) lack local dynamics, making it difficult to connect points with differing trade-offs. (3) Poor scalability of this framework to massive task scenarios. To address these issues, we adopt non-uniform rational B-Splines (NURBS) for mode connectivity, enabling the search for more intricate curves. Additionally, we introduce an order-aware objective to encourage task loss trade-offs during optimization. Moreover, we employ a task grouping strategy to enhance scalability under massive task scenarios. In a nutshell, we summarize our contributions as three-fold:

- We conduct MTL through mode connectivity by optimizing the curve connecting endpoints rather than the endpoints themselves, offering a more efficient way to leverage mode connectivity compared to PFL.
- To traverse trade-offs and approach the Pareto front, we adopt flexible NURBS curves for enhanced local adjustments. Additionally, we propose an order-aware objective to further encourage trade-off traversal. Besides, a task grouping strategy is further employed to address the scalability issue.
- Extensive experiments on key MTL datasets demonstrate that our method effectively identifies the desired point on the Pareto Frontier and achieves state-of-the-art (SOTA) performance.

## 2 Related Work

This paper primarily bridges the gap between MTL and mode connectivity, exploring their intersection. To provide context, we also introduce key developments in both domains. Additionally, we include an overview in PFL to clarify the distinctions between our approach and PFL.

### 2.1 Optimization-based MTL

Optimization-based MTL focuses on developing optimization algorithms rather than architectural changes to facilitate learning multiple tasks simultaneously. A significant body of work [Liu et al., 2024, Chen et al., 2018, Zhou et al., 2024] has explored re-weighting task losses to balance their learning progress. For instance, GradNorm defines a metric to quantify the learning progress of each task and dynamically adjusts task weights during training. Similarly, FAMO [Liu et al., 2024] refines this idea by providing a more precise estimation of progress.

Another paradigm in optimization-based MTL involves combining task gradients to address conflict and imbalance issues. MGDA [Sener and Koltun, 2018] employs the Frank–Wolfe algorithm to find the least norm combination of gradients. PCGrad [Yu et al., 2020] projects task gradients onto orthogonal directions to prevent conflicts. CAGrad [Liu et al., 2021a] balances global convergence and conflict-averse goals by designing compromise objectives. Nash-MTL [Navon et al., 2022], adopting a game-theoretic perspective, enables tasks to negotiate parameter updates for balanced progression. Recently, FairGrad [Ban and Ji, 2024] built upon Nash-MTL by introducing a finer-grained constraint to ensure balanced learning progress across tasks.

## 2.2  Mode Connectivity

Mode connectivity [Garipov et al., 2018] is a recently discovered phenomenon asserting that a low-loss path can always be found to connect two local minima. It introduces practical methods using simple curves (e.g., polygonal chains and Bézier curves) to identify such paths. Similarly, [Draxler et al., 2018] employs a nudged elastic band (NEB) method to search for low-loss connections. Expanding on this concept, SPRO[Benton et al., 2021] explores mode-connecting volumes, extending connectivity beyond simple paths. CBFT [Lubana et al., 2023] leverages mode connectivity for efficient fine-tuning in downstream tasks. Additionally, applications such as continual learning[Mirzadeh et al., 2020, Wen et al., 2023] utilize mode connectivity to transfer models along low-loss paths for specific objectives. Most existing works focus on optimizing the connected curve, which we categorize as curve-based mode connectivity for clarity.

## 2.3  Pareto Front Learning

Unlike MTL, which seeks a single optimal point on the Pareto front, Pareto front learning (PFL) aims to directly construct the entire Pareto front, allowing for user-defined trade-offs at test time. PHN [Navon et al., 2020] employs a hyper-network to generate Pareto-optimal models conditioned on preference vector inputs. In contrast, COSMOS[Ruchte and Grabocka, 2021] eliminates the parameter overhead of hyper-networks by conditioning the model on preference vectors in the feature space. Pa-MaL[Dimitriadis et al., 2023] learns the Pareto front in the manifold space by optimizing endpoints, each dedicated to a single task. To improve efficiency, subsequent works have explored techniques such as low-rank approxima-tions[Dimitriadis et al., 2024, Chen and Kwok, 2024] and mixture-of-experts (MoE) [Tang et al., 2024] to reduce

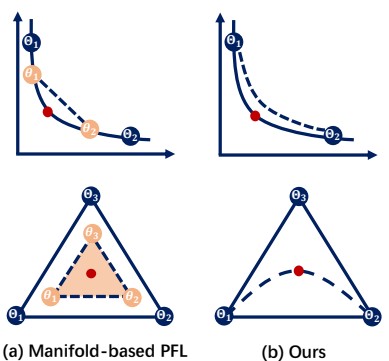

Figure 1: Comparison of manifold PFL and our work.

the parameter burden of endpoints. These manifold-based PFL approaches can be categorized as endpoint-based mode connectivity.

**Connection and Difference with Counterparts**: Inspired by mode connectivity, we aim to explore different trade-offs by aligning the low-loss path with the Pareto front, specifically for MTL rather than PFL. In other words, our focus is on obtaining desired points on the Pareto front for MTL by optimizing the curve (path), rather than constructing the entire Pareto front by optimizing endpoints. This distinction is illustrated in Figure 1 using simple two- and three-task instances. As shown, (manifold-based) PFL seeks to cover the Pareto front by optimizing endpoints and deriving the desired point through a linear combination of these endpoints. The limitations of this paradigm are discussed in Section 4.1. In contrast, our approach focuses solely on identifying the best trade-off for MTL by optimizing the curve. Moreover, our framework introduces a novel perspective for MTL, distinct from previous paradigms that rely on re-weighting task losses or gradients.

## 3  Preliminary

### 3.1  Setup of Optimization-based MTL

Optimization-based multi-task learning (MTL) methods typically assume that the model consists of a task-shared backbone network and task-specific branches. These methods aim to develop

gradient combination strategies that optimize the backbone network to benefit all tasks simultaneously. Consider a scenario with $K \geq 2$ tasks, each associated with a differentiable loss function $\mathcal{L}_i(\theta)$, where $\theta$ represents the model parameters. The objective of optimization-based MTL is to find the optimal $\theta^* \in \mathbb{R}^m$ that minimizes the losses across all tasks.

## 3.2 Pareto Concept

Let us define the weighted loss as $\mathcal{L}_\omega = \sum_{i=1}^{K} \omega_i \mathcal{L}_i(\theta)$, where $\omega \in \mathcal{W}$ and $\mathcal{W}$ represents the probability simplex over $[K]$. A point $\theta'$ is said to Pareto dominate $\theta$ if and only if $\forall i, \mathcal{L}_i(\theta') \leq \mathcal{L}_i(\theta)$. Pareto optimality occurs when no $\theta'$ exists such that $\forall i, \mathcal{L}_i(\theta') \leq \mathcal{L}_i(\theta)$. Points satisfying this condition form the Pareto set, and their corresponding solutions are called the Pareto front.

## 3.3 Mode Connectivity for MTL

For ease of explanation, we introduce the our application of vanilla mode connectivity in a 2-task MTL setting. As illustrated in Figure 2, the training process consists of two stages:
(1) First stage: Train the single-task models to obtain optimized weights, $\Theta_1$ and $\Theta_2$, each tailored for its respective task.
(2) Second stage: Initialize the endpoints with the pretrained single-task weights, $\Theta_1$ and $\Theta_2$. Using these fixed endpoints and trainable control points (e.g., $\theta_1$ and $\theta_2$), we construct a curve function $\mathcal{C}(t)$ (e.g., Bézier curve, Polygonal Chain) that connects the endpoints.

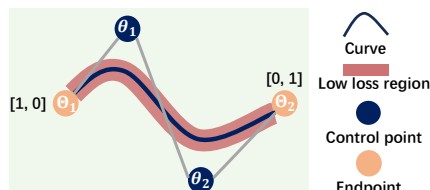

Figure 2: Illustration of mode connectivity for MTL.

During training, we uniformly sample points along the curve and minimize their corresponding losses to obtain a low-loss path:

$$\mathcal{L}_T(\theta) = \mathbb{E}_{t \sim \mathcal{U}_{[0,1]}} \left[ \mathcal{L}_1(\mathcal{C}(t; \theta)) + \mathcal{L}_2(\mathcal{C}(t; \theta)) + \lambda * \mathcal{R}_{reg} \right] \tag{1}$$

where $\mathcal{U}_{[0,1]}$ denotes the uniform distribution over the interval [0, 1], and $\mathcal{R}_{reg}$ is the (L2 norm) regularization item, and $\lambda$ is the hyper-parameter.

# 4 Motivation and Observation

Since our work is closely related to the manifold-based PFL series—which also leverages mode connectivity for multi-task learning—we draw motivation for our choice by analyzing their limitations. In addition, we highlight the challenges of using curve optimization as an alternative approach for MTL, which further motivates the designs introduced in the next section.

## 4.1 Limitations of Manifold-based PFL

To facilitate a clear discussion, we categorize the scenarios into two-task and multi-task settings. We adopt PaMaL as the baseline for our experiments, since most existing manifold-based PFL approaches that utilize mode connectivity are either derived from or closely related to PaMaL.

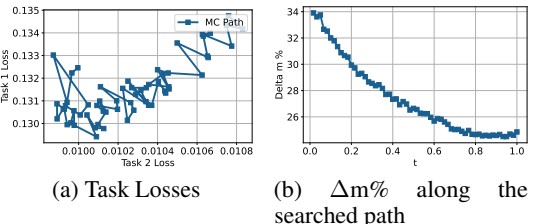

(a) Task Losses    (b) $\Delta$m% along the searched path

Figure 3: The results of PaMaL evaluated on CityScapes. (a) is the task losses while (b) is the $\Delta$m% along the searched path (MC path).

**Degeneration of 2-Task Scenario**: In the 2-task scenario, PaMaL reduces to linear mode connectivity, as illustrated in Figure 1. In other words, regardless of how the models (endpoints) are trained, their final weighted combination, $t * \theta_1 + (1 - t) * \theta_2$, lies along a linear path. However, it is generally challenging to approximate the Pareto front using a linear path. To verify this, we re-implemented PaMaL and evaluated it on the CityScapes dataset, presenting the results in Figure 3. As shown in Figure 3(a), task losses along the path fall within a narrower range, forming a relatively flat region rather than a well-defined Pareto front. Additionally, Figure 3(b) illustrates the changes in $\Delta$m% (defined in Eqn. 8) along the path. Notably, $\Delta$m% exhibits a monotonically decreasing trend, which deviates from the expected characteristics of the Pareto front.

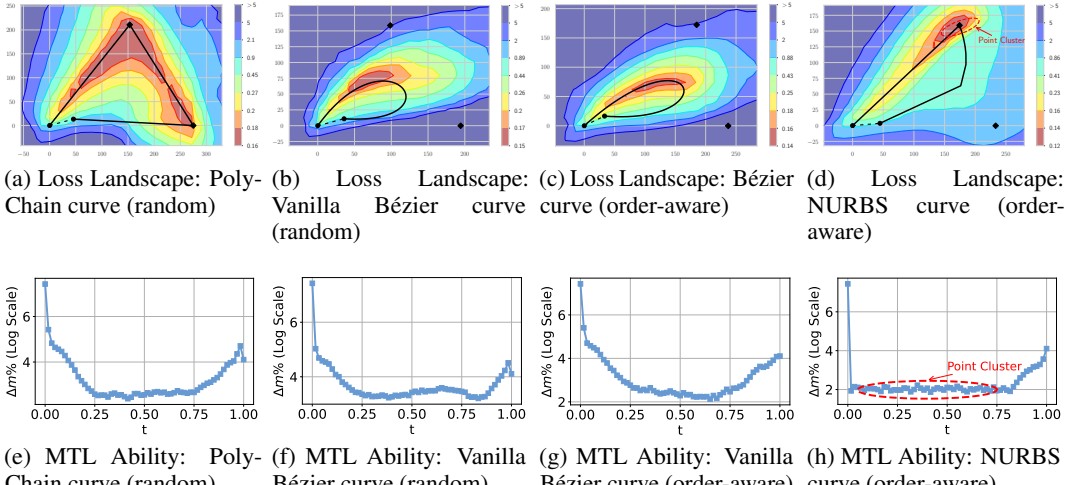

(a) Loss Landscape: Poly-Chain curve (random)  (b) Loss Landscape: Vanilla Bézier curve (random)  (c) Loss Landscape: Bézier curve (order-aware)  (d) Loss Landscape: NURBS curve (order-aware)

(e) MTL Ability: Poly-Chain curve (random)  (f) MTL Ability: Vanilla Bézier curve (random)  (g) MTL Ability: Vanilla Bézier curve (order-aware)  (h) MTL Ability: NURBS curve (order-aware)

Figure 4: The results of employing PolyChain, Bézier and NURBS curve for mode connectivity on CityScapes. (a)-(d) are the loss landscapes (average task loss) and searched curves with respect to different curves and objectives. • and ♦ are endpoint and control point, respectively, while − is the searched curve. All endpoints are well trained single task weights. (e)-(h) are the mtl ability ($\Delta \mathbf{m\%}$) evaluation along the corresponding searched curves.

**Scaling Problem of Many Task Scenario**: Since manifold-based PFL optimizes endpoints to approximate the Pareto front, the number of endpoints must scale with the number of tasks. Consequently, in scenarios involving many tasks (>2), this approach demands significant computational and storage resources, making it challenging to optimize. We present the memory cost as the number of tasks increases in Figure 5(a). As expected, PaMaL exhibits an almost linear increase in memory cost with respect to the number of tasks [2]. While subsequent methods, e.g., PaLoRA [Dimitriadis et al., 2024], employ techniques like LoRA or MoE to address these scalability issues, they still encounter optimization difficulties. Despite introducing additional parameters, their performance often lags behind advanced MTL approaches. As illustrated in Figure 5(b), PaMaL and its efficient variant, PaLoRA, achieve SOTA performance on the CityScapes dataset (2-task scenario). However, they are less competitive on NYUv2 (3-task scenario), demonstrating the difficulty of such manifold-based optimization [3].

## 4.2 Challenges of Curve-Based Mode Connectivity

To circumvent these issues intrinsic lie in the endpoint-based mode connectivity, we opt to the curve-based ones. However, it will naturally encounters the following challenges:

### 4.2.1 Challenge 1: Randomness

The original mode connectivity approach seeks a low-loss curve (e.g., Bézier or Polygonal Chain) by minimizing task losses but lacks designs specifically tailored for handling multi-task competition. As a result, its effectiveness in MTL exhibits randomness.

We verify this by visualizing the curve on the loss landscape and computing its MTL ability, quantified by $\Delta m\%$ (defined in Eqn.8), along different points of the curve, as shown in Figure 4(a)(e)-(b)(f). Although most points along

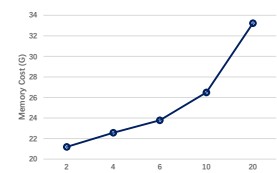

(a) Memory cost of PaMaL

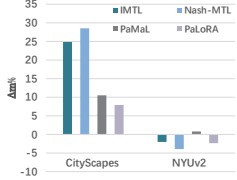

(b) Performance comparison.

Figure 5: Analysis on PaMaL. (a) is the memory cost of PaMaL with the increasing number of endpoints. (b) is the performance compared with advanced MTL on CityScapes and NYUv2 datasets.

---

[2]More memory cost comparisons are presented in the **Appendix** (Sec. B.3).

[3]We calculate $\Delta m\%$ according to the results provided in PaLoRA.

the curve lie within a low-loss region, their MTL abilities fluctuate significantly, indicating a failure to align with the Pareto front. This suggests that mode connectivity alone does not inherently promote stable trade-offs and may struggle to identify the optimal trade-off point (i.e., the best MTL ability) along the curve.

To address this randomness issue, we introduce an additional objective (see Section 5.1) to promote a more structured exploration of trade-offs. Figure 4(c)(g) provides a preview of the benefits of our approach. As shown, the proposed objective guides the curve through a broader low-loss region and successfully identifies a better and more distinct optimal trade-off compared to Figure 4(a)(e)-(b)(f).

### 4.2.2 Challenge 2: Flexibility

Another challenge concerns the choice of curve. The commonly used Bézier curve in mode connectivity is often too smooth [Draxler et al., 2018] to effectively explore diverse trade-offs. As shown in Figure 4(b)(c), each control point exerts a global influence on the curve, limiting its ability to capture local dynamics. In contrast, NURBS offers greater flexibility for local adjustment through segmental modeling. As illustrated in Figure 4(d), even with uniformly sampled points, most lie within a low-loss basin (Point Cluster) and demonstrate strong MTL performance, underscoring NURBS's advantage in modeling local variations.

### 4.2.3 Challenge 3: Scalability

The last challenge arises in terms of scalability when addressing scenarios involving a large number of tasks. As shown in Figure 2, curve-based mode connectivity is primarily applicable to settings involving two endpoints (tasks), which limits its utility in massive task scenarios—a critical aspect of MTL. Although some prior works have attempted to extend mode connectivity to multiple endpoints [Fort and Jastrzebski, 2019] or adopt manifold-based strategies [Benton et al., 2021], these approaches are often difficult to optimize or require extensive pre-training of multiple endpoints. These limitations are key reasons why we do not adopt endpoint-based mode connectivity in our framework.

**Remarks:** This scalability issue is shared by both curve- and endpoint-based mode connectivity approaches. However, they pursue different goals: endpoint-based mode connectivity aims to approximate the entire Pareto front, while our curve-based approach focuses on exploring the optimal trade-off point for MTL. This key difference allows us to apply endpoint clustering when facing massive tasks—a strategy that is not feasible for the endpoint-based approach.

## 5 Principal Design

### 5.1 Injecting Order to Mode Connectivity

To address the *Challenge 1*, we begin by recalling a key property of the Pareto front: the ability to traverse trade-offs. Building on this property, we propose the following order-aware objective. Assume we sample two variables, $t_1$ and $t_2$, and obtain the corresponding weights, $\mathcal{C}(t_1)$ and $\mathcal{C}(t_2)$. The objective is then defined as:

$$\mathcal{R}_o = \frac{1}{2} \left[ e^{[\mathcal{L}_1(t_1) - \mathcal{L}_1(t_2)](t_2 - t_1)} + e^{[\mathcal{L}_2(t_1) - \mathcal{L}_2(t_2)](t_1 - t_2)} \right] \tag{2}$$

where $\mathcal{L}_i(t_j)$ is the abbreviate of $\mathcal{L}_i(\mathcal{C}(t_j; \theta))$, and represent the losses of task $i$ at $\mathcal{C}(t_j; \boldsymbol{\theta})$. This constraint explicitly encourages the task losses to exhibit different tendencies with respect to $t$. The overall objective can then be defined as follows:

$$\mathcal{J}(\theta) = \mathcal{L}_T + \alpha * \mathcal{R}_o \tag{3}$$

where $\mathcal{L}_T$ represents the mean task losses, and $\alpha$ is a hyper-parameter. This objective ensures the low-loss property while encouraging the searched curve to gradually approach the Pareto front.

### 5.2 Enabling Complicated Path: NURBS

As *Challenge 2* have mentioned, Bézier curves often suffer from an over-smoothing issue, making it challenging to capture long-distance trade-offs. This limitation arises because the order of the curve

is inherently tied to the number of control points, causing adjustments to any control point to shift the entire curve [Piegl and Tiller, 2012]. To provide the curve with greater flexibility, we adopt NURBS, which decouples the number of control points (knots) from the curve order. This approach enables more localized dynamics without significantly affecting the global structure of the curve. Formally, the adopted NURBS equation is defined as follows:

$$\mathcal{C}(t) = \sum_{i=1}^{K} \left( \frac{N_{i,p}(t)\, w_i}{\sum_{i=1}^{K} N_{i,p}(t)\, w_i} \right) \theta_i \tag{4}$$

where $N_{i,p}(t)$ represents the $p$-th B-spline basis function, and $\theta_i$ is the $i$-th control point (knot), while $w_i$ is the learnable weight factor. And the basis function $N_{i,p}(t)$ is derived using the Cox-de Boor recurrence formula [Boor, 1971], as follows:

$$N_{i,1}(t) = \begin{cases} 1 & t_i \leq t < t_{i+1}, \\ 0 & \text{otherwise} \end{cases} \tag{5}$$

$$N_{i,n}(t) = \frac{t - t_i}{t_{i+n-1} - t_i} N_{i,n-1}(t) + \frac{t_{i+n} - t}{t_{i+n} - t_{i+1}} N_{i+1,n-1}(t)$$

These segmentations of $t$ enable the NURBS curve to capture local dynamics without significantly altering the entire curve, making it more effective for exploring the complex loss landscape of this task. We also present the corresponding visualization results of employing NURBS in Figure 4(d)-(h). Compared to the Poly Chain and Bézier curves, NURBS exhibits greater flexibility, a longer low-loss path, and improved MTL performance.

### 5.3 Scale to Many Task Scenarios

Equations 2 and 3 are specifically designed for two-task scenarios, facilitating trade-offs between the two tasks. To address *Challenge 3*, we propose a warmup-based task grouping strategy that scales to many-task settings by dividing tasks into three clusters.

Prior to the main training phase, we perform a warmup stage for $e$ epochs to collect gradient statistics. For a mini-batch, we obtain the set of task gradients $\{g_1, ..., g_K\}$. Based on these, we compute a cosine similarity matrix $A$: $A_{ij} = \frac{g_i^\top g_j}{|g_i||g_j|}$. Using $A$, we construct a graph $\mathcal{G} = (\mathcal{V}, \mathcal{E})$, where each node $v_i \in \mathcal{V}$ represents a task and each edge $e_{ij} \in \mathcal{E}$ corresponds to $A_{ij}$ for $i < j \leq K$. We then apply spectral clustering [Ng et al., 2001] to partition $\mathcal{G}$ into three clusters $\mathcal{P} = \{\mathcal{E}_1, \mathcal{E}_2, \mathcal{E}_3\}$.

After pretraining on clusters $\mathcal{E}_1$ and $\mathcal{E}_2$, we obtain their respective weights $\Theta_1$ and $\Theta_2$, which are used to initialize the endpoints of the connectivity curve. To align the trade-offs across all clusters, we introduce the following alignment objective:

$$\mathcal{R}_{align} = e^{-|t - \frac{1}{2}|} \cdot \mathcal{L}_r \tag{6}$$

where $\mathcal{L}_r$ denotes the average loss of tasks in cluster $\mathcal{E}_3$. The full training objective is then defined as:

$$\mathcal{J}(\theta) = \mathcal{L}_T + \alpha \cdot \mathcal{R}_o + \mathcal{R}_{align} \tag{7}$$

This formulation ensures that $\mathcal{L}_r$ reaches its minimum at $t = 1/2$ while maintaining mode connectivity between $\mathcal{E}_1$ and $\mathcal{E}_2$. This enables aligned trade-offs across all task clusters at the center of the curve. Note that $K = 3$ is a special case where grouping is not required.

## 6    Performance Evaluation

In this section, we first evaluate our method using mainstream MTL benchmarks and compare it with the following baselines: Linear Scalarization (LS), Scale-Invariant (SI), RLW [Lin et al., 2021], DWA [Liu et al., 2019], UW [Kendall et al., 2018], MGDA [Sener and Koltun, 2018], GradDrop [Chen et al., 2020], PCGrad [Yu et al., 2020], CAGrad [Liu et al., 2021a], IMTL [Liu et al., 2021b], Nash-MTL [Navon et al., 2022], FAMO [Liu et al., 2024], and FairGrad [Ban and Ji, 2024]. We also compare with several PFL approaches, including COSMOS [Ruchte and Grabocka, 2021], HPN [Navon et al., 2020], and PaMaL [Dimitriadis et al., 2023], to demonstrate EXTRA's ability to approach the Pareto front. Additionally, we provide further analysis, e.g., ablation study

Table 2: **Scene understanding** (*NYUv2*, 3 tasks). We report MTAN model performance averaged over 3 random seeds. The best scores are provided in gray, and the second scores are underlined.

| Method | Segmentation ↑ | | Depth ↓ | | Surface Normal | | | | | MR ↓ | Δm% ↓ |
|---|---|---|---|---|---|---|---|---|---|---|---|
| | | | | | Angle Distance ↓ | | Within $t°$ ↑ | | | | |
| | mIoU | Pix. Acc. | Abs. Err. | Rel. Err. | Mean | Median | 11.25 | 22.5 | 30 | | |
| Independent | 38.30 | 63.76 | 0.68 | 0.28 | 25.01 | 19.21 | 30.14 | 57.20 | 69.15 | - | - |
| LS | 39.29 | 65.33 | 0.55 | 0.23 | 28.15 | 23.96 | 22.09 | 47.50 | 61.08 | 10.67 | 5.46 |
| RLW | 37.17 | 63.77 | 0.58 | 0.24 | 28.27 | 24.18 | 22.26 | 47.05 | 60.62 | 13.22 | 7.67 |
| DWA | 39.11 | 65.31 | 0.55 | 0.23 | 27.61 | 23.18 | 24.17 | 50.18 | 62.39 | 9.78 | 3.49 |
| Uncertainty | 36.87 | 63.17 | 0.54 | 0.23 | 27.04 | 22.61 | 23.54 | 49.05 | 63.65 | 9.67 | 4.01 |
| MGDA | 30.47 | 59.90 | 0.61 | 0.26 | 24.88 | 19.45 | 29.18 | 56.88 | 69.36 | 7.56 | 1.47 |
| GradDrop | 39.39 | 65.12 | 0.55 | 0.23 | 27.48 | 22.96 | 23.38 | 49.44 | 62.87 | 10.11 | 3.61 |
| PCGrad | 38.06 | 64.64 | 0.56 | 0.23 | 27.41 | 22.80 | 23.86 | 49.83 | 63.14 | 10.33 | 3.83 |
| CAGrad | 39.79 | 65.49 | 0.55 | 0.23 | 26.31 | 21.58 | 25.61 | 52.36 | 65.58 | 7.56 | 0.29 |
| IMTL | 39.35 | 65.60 | 0.54 | 0.23 | 26.02 | 21.19 | 26.20 | 53.13 | 66.24 | 6.67 | -0.59 |
| Nash-MTL | 40.13 | 65.93 | 0.53 | 0.22 | 25.26 | 20.08 | 28.40 | 55.47 | 68.15 | 4.11 | -4.04 |
| FAMO | 40.30 | 66.07 | 0.56 | 0.21 | 26.67 | 21.83 | 25.61 | 51.78 | 64.85 | 6.56 | 0.16 |
| FairGrad | 39.74 | 66.01 | 0.54 | 0.22 | 24.84 | 19.60 | 29.26 | 56.58 | 69.16 | 3.56 | -4.66 |
| EXTRA-L | 40.90 | 66.67 | 0.54 | 0.22 | 25.16 | 19.88 | 28.79 | 55.94 | 68.50 | 3.67 | $-4.41_{\pm1.3}$ |
| EXTRA-F | 41.89 | 67.60 | 0.53 | 0.22 | 24.64 | 19.30 | 29.66 | 57.30 | 69.76 | 1.56 | $-6.30_{\pm1.2}$ |

(Sec. B.1), grouping strategy comparison (Sec. B.2), plug-and-play verification (Sec. B.4), analysis on control point number (Sec. B.5), and hyper-parameter analysis (Sec. B.6), in **Appendix** to offer deeper insights into the performance. All experiments were conducted on Tesla V100 GPUs. Please refer to **Appendix** (Sec. A) for more implementation details.

**Evaluation Metric**. In addition to reporting individual performance, we also incorporate a widely used metric, Δm% [Maninis et al., 2019], which evaluates the overall degradation compared to independently trained models that are considered as the reference oracles. The formal definition of Δm% is given as:

$$\Delta\mathbf{m}\% = \frac{1}{K}\sum_{k=1}^{K}(-1)^{\delta_k}(M_{m,k} - M_{b,k})/M_{b,k} \times 100 \tag{8}$$

where $M_{m,k}$ and $M_{b,k}$ represent the metric $M_k$ for the compared method and the independent model, respectively. The value of $\delta_k$ is assigned as 1 if a higher value is better for $M_k$, and 0 otherwise. Besides, we also report another popular metric named **Mean Rank** (**MR**), which computes the average ranks of each methods across all tasks.

## 6.1 Overall MTL Evaluation

In this section, we evaluate our method on two image classification tasks, CelebA and MultiMNIST, as well as two scene understanding datasets, CityScapes and NYUv2. CelebA serves as a multi-task dataset to assess the model's ability to handle a large number of tasks, while MultiMNIST is a toy dataset used to demonstrate the model's ability to cover the Pareto front. Notably, 'EXTRA-L' represents equips with LS trained endpoints, while 'EXTRA-F' represents equips with both LS and FairGrad trained endpoints.

### 6.1.1 Scene Understanding

**CityScapes**: CityScapes is a large-scale dataset designed for scene understanding, which we use as an MTL benchmark focusing on the tasks of semantic segmentation and depth estimation. The results, presented in Table 3, demonstrate that EXTRA achieves SOTA performance based on Δm%, a key metric widely regarded as one of the most important in

Table 1: Results on *CelebA*.

| Method | CelebA | |
|---|---|---|
| | MR ↓ | Δm% ↓ |
| LS | 8.00 | 4.15 |
| SI | 9.75 | 7.20 |
| RLW | 6.80 | 1.46 |
| DWA | 8.88 | 2.40 |
| UW | 7.48 | 3.23 |
| MGDA | 12.95 | 14.85 |
| PCGrad | 8.60 | 3.17 |
| CAGrad | 8.05 | 2.48 |
| IMTL-G | 6.45 | 0.84 |
| Nash-MTL | 6.53 | 2.84 |
| FAMO | 6.35 | 1.21 |
| FairGrad | 7.05 | 1.15 |
| EXTRA-L | 4.15 | $0.10_{\pm0.5}$ |
| EXTRA-F | 3.98 | $-0.11_{\pm0.2}$ |

MTL. Additionally, it ranks highly in terms of the **MR** metric, which evaluates the ranking position of `EXTRA` among various MTL approaches. Notably, our method is competitive with advanced gradient-based MTL approaches, highlighting that our novel framework is as effective as the alternative optimization trajectory paradigm and offers a fresh perspective for advancing MTL research.

**NYUv2**: NYUv2 is another indoor scene understanding dataset commonly used for MTL benchmarking, comprising three tasks: semantic segmentation, depth estimation, and surface normal prediction. We conducted experiments on this dataset, and the results are presented in Table 2. As shown, `EXTRA` significantly outperforms its counterparts in both $\Delta$m% and **MR**. Moreover, it achieves top performance on nearly every single task, demonstrating `EXTRA`'s capability to search for the optimal trade-offs. We attribute this effectiveness to our design, which facilitates traversing trade-offs while maintaining low losses along the searched path. Even `EXTRA-L` exhibits competitive performance though their endpoints are naïvely trained with LS.

Table 3: **Scene understanding** (*CityScapes*, 2 tasks).

| Method | Segmentation ↑ | | Depth ↓ | | MR ↓ | Δm% ↓ |
|---|---|---|---|---|---|---|
| | mIoU | Pix. Acc. | Abs. Err. | Rel. Err. | | |
| Independent | 74.01 | 93.16 | 0.0125 | 27.77 | - | - |
| LS | 75.18 | 93.49 | 0.0155 | 46.77 | 9.25 | 22.60 |
| RLW | 74.57 | 93.41 | 0.0158 | 47.79 | 12.00 | 24.37 |
| DWA | 75.24 | 93.52 | 0.0160 | 44.37 | 9.25 | 21.43 |
| Uncertainty | 72.02 | 92.85 | 0.0140 | 30.13 | 8.25 | 5.88 |
| MGDA | 68.84 | 91.54 | 0.0309 | 33.50 | 12.00 | 44.14 |
| GradDrop | 75.27 | 93.53 | 0.0157 | 47.54 | 8.75 | 23.67 |
| PCGrad | 75.13 | 93.48 | 0.0154 | 42.07 | 9.50 | 18.21 |
| CAGrad | 75.16 | 93.48 | 0.0141 | 37.60 | 8.50 | 11.58 |
| IMTL | 75.33 | 93.49 | 0.0135 | 38.41 | 6.75 | 11.04 |
| Nash-MTL | 75.41 | 93.66 | 0.0129 | 35.02 | 4.00 | 6.82 |
| FAMO | 74.54 | 93.29 | 0.0145 | 32.59 | 9.00 | 8.13 |
| FairGrad | 75.72 | 93.68 | 0.0134 | 32.25 | 2.50 | 5.18 |
| `EXTRA-L` | 75.53 | 93.63 | 0.0127 | 33.45 | 3.25 | $4.93_{\pm 1.1}$ |
| `EXTRA-F` | 76.11 | 93.58 | 0.0126 | 30.20 | 2.00 | $1.63_{\pm 1.6}$ |

### 6.1.2 Image Classification

**CelebA**: CelebA [Liu et al., 2015] is a widely used facial attributes dataset containing over 200,000 images annotated with 40 binary attributes. Recently, it has been adopted as a 40-task multi-task learning (MTL) benchmark to assess a model's capacity to handle a large number of tasks. Following the experimental setup of prior work [Liu et al., 2024, Ban and Ji, 2024], we evaluate `EXTRA` on CelebA, with results presented in Table 1. FairGrad results are reported from three random runs using the same seeds as `EXTRA`, based on its official implementation. As shown, `EXTRA` achieves SOTA performance in both $\Delta$m% and **MR**. These results indicate that `EXTRA` can enhance all individual tasks simultaneously and underscore the effectiveness of our grouping strategy in large-scale task scenarios.

**MultiMNIST**: Following the experimental setup in Pa-MaL [Dimitriadis et al., 2023], we evaluate `EXTRA` on MultiMNIST, a widely used dataset for assessing how MTL/PFL methods approach the Pareto front. The results are presented in Figure 6. As shown, the best trade-off identified by `EXTRA` is competitive with SOTA MTL methods. Additionally, `EXTRA` achieves broad coverage of trade-offs without sacrificing performance, demonstrating its superior ability to approach the Pareto front compared to its counterparts in the 2-task scenario [4].

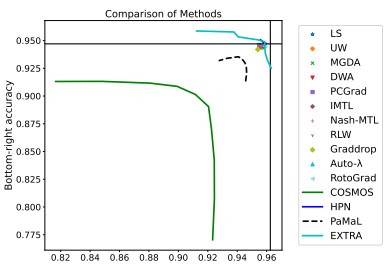

Figure 6: Results on *MultiMNIST*.

## 7   Conclusion

In this paper, we approach MTL from the perspective of mode connectivity, in contrast to traditional gradient-based methods. Specifically, our focus is on exploring the optimal trade-offs that deliver competitive MTL performance, rather than aiming to directly approach the Pareto front. By designing an order-aware objective and employing a more flexible curve (NURBS), we show that optimizing the curve, rather than just the endpoints, is both more efficient and effective for MTL. This approach

---

[4]We report the results of PFL approaches based on runs of PaMaL's official implementation, while MTL results are taken from PaMaL's paper.

is empirically validated across multiple mainstream datasets. We hope that our work offers valuable insights for future research in the field of MTL.

## Acknowledgement

This research is supported, in part, by the WeBank-NTU Joint Research Institute on Fintech, Jinan-NTU Green Technology Research Institute (GreenTRI), and Joint NTU-UBC Research Centre of Excellence in Active Living for the Elderly (LILY), Nanyang Technological University, Singapore. It's also supported, in part, by the NTU-PKU Joint Research Institute, a collaboration between Nanyang Technological University and Peking University that is sponsored by a donation from the Ng Teng Fong Charitable Foundation. The first author also would like to thank the insightful discussion with Dr. Shuaicheng Niu, Shibo Feng, and Haochen Li.

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

# A   Implementation Details

## A.1   Evaluation Protocol

For scene understanding benchmarks such as CityScapes and NYUv2, mainstream multi-task learning (MTL) approaches typically report the final performance by averaging results over the last 10 epochs, due to the absence of a validation set. However, this evaluation protocol is not directly applicable to our method, which generates a solution curve during training and evaluates sampled points along the curve to demonstrate superior trade-offs. Specifically, we save the trained model at the final epoch and evaluate MTL performance at $t = \frac{1}{2}$, which represents the midpoint of the curve.

For image classification benchmarks such as CelebA, we adopt the same training procedure as used for the scene understanding benchmarks. However, we utilize a validation set to select the optimal value of $t$ and report the corresponding MTL performance.

## A.2   Experimental Setting

In line with the implementation and training strategy of FairGrad [Ban and Ji, 2024], we construct our model using SegNet [Badrinarayanan et al., 2017], with MTAN [Liu et al., 2019] employed as the backbone. The model is trained using the Adam optimizer for 200 epochs. The initial learning rate is set to 1e-4 and decayed by a factor of 2 after 100 epochs. The batch size is set to 2 for NYUv2 and 8 for CityScapes. The control point numbers are 4 and 5 on CityScapes and NYUv2, respectively, with 2 and 3 trainable in the second stage.

For the CelebA dataset, we adopt a 9-layer convolutional neural network (CNN) as the backbone, with task-specific linear heads appended. The model is trained with the Adam optimizer for 15 epochs using an initial learning rate of 3e-4 and a batch size of 256. The control point number is 3, with 1 trainable in the second stage.

Regarding the MultiMNIST dataset, we follow the protocol described in PaMaL [Dimitriadis et al., 2023]. Each MultiMNIST image is formed by sampling (with replacement) two MNIST digits (28×28), which are placed at the top-left and bottom-right of a 36×36 grid. This composite image is then resized to 28×28 pixels. The resulting dataset comprises 60,000 training, 10,000 validation, and 10,000 test samples. The model uses a LeNet-style shared-bottom architecture: the encoder contains two convolutional layers with 10 and 20 channels (kernel size 5), each followed by max pooling and ReLU activation. The encoder outputs a 50-dimensional embedding. Each decoder consists of two fully connected layers, with the final output layer producing predictions over 10 classes. The model is trained using Adam with a learning rate of 0.001, no learning rate scheduler, a batch size of 256, and a total of 10 training epochs. The control point number is 3, with 1 trainable in the second stage.

# B   Additional Experiments

## B.1   Ablation Study

Our system comprises multiple components, including the order-aware objective ($\mathcal{R}_o$), alignment objective ($\mathcal{R}_{\text{align}}$ in Eqn.7), and curve selection. To evaluate the effectiveness and rationale behind each component, we conduct an ablation study on the NYUv2 dataset, with the results presented in Table 4. As shown, without $\mathcal{R}_{\text{align}}$, EXTRA still outperforms the baseline but falls short of the complete system's performance. This suggests that, while the model can minimize loss, it fails to properly calibrate the remaining tasks at $t = \frac{1}{2}$. Additionally, EXTRA shows slight improvements without $\mathcal{R}_o$, demonstrating the strong capability of NURBS in capturing diverse trade-offs. This observation is further corroborated by the comparison between NURBS and Bézier, which reveals a substantial performance gap in MTL.

## B.2   Grouping Strategy Comparison

To further demonstrate the effectiveness of the proposed grouping strategy, we compare it with two alternative approaches: random grouping and K-means clustering. The corresponding results are shown in Figure 7. In this experiment, the 40 tasks are grouped into 3 clusters using each of the three strategies, followed by multi-task learning (MTL) training. As illustrated, our strategy achieves

Table 4: Ablation study of EXTRA on *NYUv2* (3 tasks).

| $\mathcal{R}_o$ | $\mathcal{R}_{align}$ | Bézier | NURBS | $\Delta\mathbf{m}\%\downarrow$ |
|---|---|---|---|---|
| | | | | -4.66 |
| ✓ | | | ✓ | -6.05 |
| | ✓ | | ✓ | -4.97 |
| ✓ | ✓ | ✓ | | -4.36 |
| ✓ | ✓ | | ✓ | -6.30 |

the best overall MTL performance, outperforming both random and K-means grouping. Notably, K-means clustering fails to deliver satisfactory results due to imbalanced cluster sizes. Specifically, we observe that two of the three K-means clusters contain only a single task, which hinders the exploration of trade-offs under our training framework.

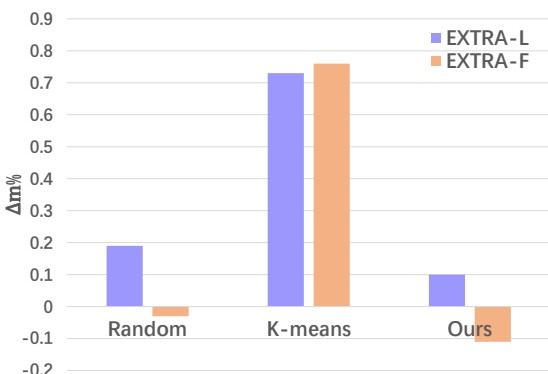

Figure 7: Comparison of grouping strategies on CelebA. 'Random' represents the uniform division for 40 tasks, while K-means represents leveraging K-means to cluster the warmup gradients of 40 tasks.

## B.3 Memory Cost Analysis

To evaluate the efficiency and scalability of our method compared to endpoint-based approaches, we statistically analyze their memory consumption during training across various MTL benchmarks, as shown in Figure 8. As depicted, in the 2-task (CityScapes) and 3-task (NYUv2) settings, EXTRA incurs slightly higher memory usage due to its two-stage training paradigm. However, in the large-scale scenario with 40 tasks (CelebA), endpoint-based methods such as PaMaL require substantial memory resources, resulting in an out-of-memory (OOM) issue during training. In contrast, EXTRA maintains a manageable memory footprint, demonstrating better scalability.

## B.4 Plug-and-Play Verification

In addition to LS and FairGrad, we further incorporate another mainstream MTL approach, CAGrad, to demonstrate the plug-and-play capability of EXTRA, with the corresponding results shown in Table 5. As illustrated, EXTRA also provides significant improvements when applied to CAGrad, following the same trend observed with LS and FairGrad. Specifically, EXTRA not only enhances overall MTL performance but also improves each individual metric, thereby verifying its plug-and-play effectiveness.

## B.5 Analysis on Control Point Number

We further analyze the effect of the number of bends in the NURBS representation on the CityScapes dataset, with results presented in Figure 10. As shown, EXTRA-L achieves the best performance in terms of $\Delta$m% when the number of bends is set to 4. Notably, reducing the number of bends to 3 leads to a substantial drop in performance, likely due to the limited expressive capacity of the

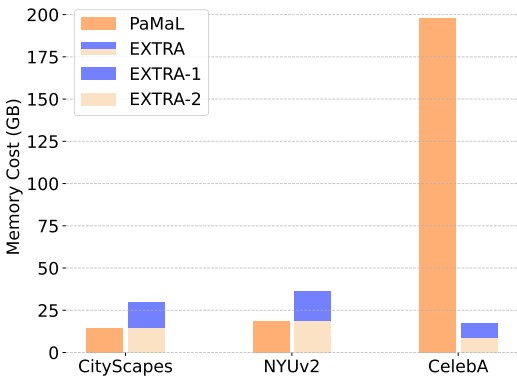

Figure 8: Memory Cost Comparison. 'EXTRA-1' and 'EXTRA-2' represents the first and second training stage of EXTRA. Note that the memory consumption of PaMaL on the CelebA dataset is estimated, as its actual training raises an out-of-memory (OOM) issue under our experimental settings.

Table 5: **Scene understanding** (*CityScapes*, 2 tasks). ▲/▼ indicates outperforms/underperforms their vanilla versions. 'EXTRA-L', 'EXTRA-C', and 'EXTRA-F' are EXTRA augmented LS, CAGrad, and FairGrad versions.

| Method | Segmentation ↑ | | Depth ↓ | | $\Delta\mathbf{m}\%$ ↓ |
|---|---|---|---|---|---|
| | mIoU | Pix. Acc. | Abs. Err. | Rel. Err. | |
| Independent | 74.01 | 93.16 | 0.0125 | 27.77 | - |
| LS | 75.18 | 93.49 | 0.0155 | 46.77 | 22.60 |
| EXTRA-L | 75.53 ▲ | 93.63 ▲ | 0.0127 ▲ | 33.45 ▲ | 4.93 ▲ |
| CAGrad | 75.16 | 93.48 | 0.0141 | 37.60 | 11.58 |
| EXTRA-C | 75.50 ▲ | 93.55 ▲ | 0.0135 ▲ | 35.73 ▲ | 8.61 ▲ |
| FairGrad | 75.72 | 93.68 | 0.0134 | 32.25 | 5.18 |
| EXTRA-F | 76.11 ▲ | 93.58 ▼ | 0.0126 ▲ | 30.20 ▲ | 1.63 ▲ |

NURBS curve. Conversely, increasing the number of bends to 5 or 6 does not yield performance gains, which is somewhat counterintuitive. To investigate this further, we visualize the corresponding loss landscapes in Figure 9. As illustrated, EXTRA-3 degenerates into a Bézier-like curve, exhibiting excessive smoothness. Meanwhile, the curves produced by EXTRA-5 and EXTRA-6 appear irregular and less stable, likely due to the increased difficulty in optimization, which in turns highlights the superiority of choosing curve-based rather than endpoint-based mode connectivity for MTL.

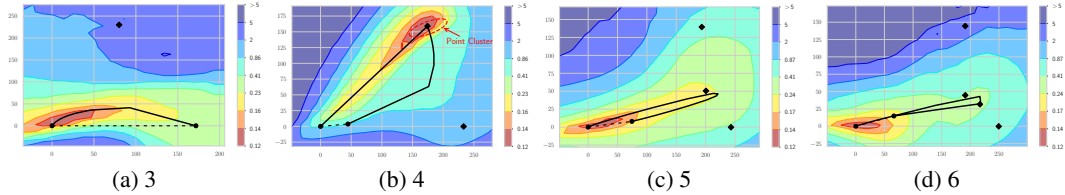

(a) 3      (b) 4      (c) 5      (d) 6

Figure 9: The loss landscape of employing NURBS with different control point number (including endpoints). We abbreviate them as EXTRA-3, EXTRA-4, EXTRA-5, and EXTRA-6.

## B.6 Hyper-parameter Analysis

We evaluate the impact of the hyperparameter $\alpha$ on the final performance and present the results in Figure 11. As shown, the performance remains relatively stable across different values of $\alpha$, with the highest average performance achieved at $\alpha = 0.3$. However, $\alpha = 0.5$ offers a better trade-off between average performance and variance. Therefore, we adopt $\alpha = 0.5$ as the default setting in our experiments.

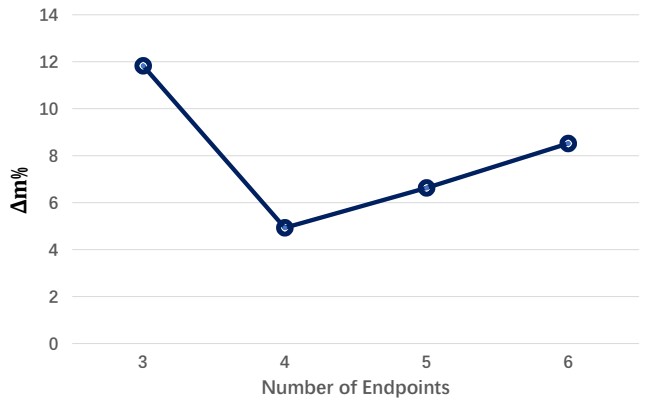

Figure 10: Analysis of number of bends on *CityScapes*, 2 tasks.

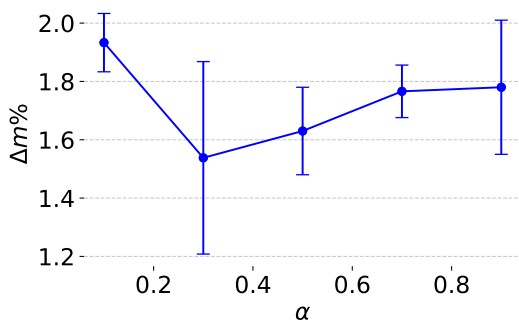

Figure 11: Analysis on $\alpha$.

## C  Limitation and Discussion

Although EXTRA achieves state-of-the-art performance on mainstream MTL benchmarks, it faces challenges in delivering user-preference MTL results when the number of tasks exceeds two. While a Bézier surface can align mode connectivity with the Pareto front for three tasks, this approach becomes progressively more difficult as the number of tasks increases, which represents a key limitation of our method. Addressing this limitation will be a focal point of future work. Additionally, one might raise concerns regarding the fairness of our evaluation, given that EXTRA utilizes multiple points (endpoints and control points) to construct the curve, thereby increasing the model's capacity. While we acknowledge this concern, we emphasize that our work introduces a novel perspective to MTL, potentially offering an alternative to gradient-based MTL approaches, which have been debated for their effectiveness [Xin et al., 2022].

