# OpenReview forum: "Exploring Tradeoffs through Mode Connectivity for Multi-Task Learning"
_NeurIPS.cc/2025/Conference — NeurIPS 2025 poster_

### Official Review · Reviewer_5udz · 2025-06-11

**Clarity:** 2
**Significance:** 2
**Originality:** 3
**Rating:** 4
**Confidence:** 4

**Summary:**

The paper introduces EXTRA, a method that explores optimal trade-offs in multi-task learning by learning a parameter-space curve connecting task-specific models, such that all points along the curve achieve low losses across tasks. Unlike Pareto Front Learning approaches that optimize multiple endpoints, EXTRA focuses on the loss-loss path itself, enabling efficient discovery of Pareto-optimal solutions. The authors identify that Bézier curves lack local flexibility, making them insufficient for modeling nuanced trade-offs in deep loss landscapes. Separately, they highlight that endpoint-based methods suffer from poor scalability as the number of tasks grows. To address both issues, EXTRA employs non-uniform rational B-splines (NURBS) for mode connectivity, which provide finer local control over the curve while keeping memory usage efficient. Experiments on diverse multi-task benchmarks demonstrate that EXTRA achieves state-of-the-art performance and remains compatible with existing MTL methods.

**Questions:**

Please refer to the weaknesses section of the paper. I’m particularly interested in how the reviewers will respond to these points.

**Ethical Concerns:**

["NO or VERY MINOR ethics concerns only"]

**Final Justification:**

The paper’s contributions merit acceptance, provided the authors address my concerns in a future revision.

**Limitations:**

yes

**Paper Formatting Concerns:**

No major formatting issues observed.

**Quality:**

3

**Strengths And Weaknesses:**

Strengths
1. Novel use of mode connectivity for MTL Optimization: Unlike prior Pareto Front Learning (PFL) approaches that optimize multiple endpoints, EXTRA leverages mode connectivity by directly optimizing the path between endpoints, enabling efficient discovery of optimal trade-off points.
2. Increased flexibility through NURBS curves: The paper addresses the global smoothness limitation of traditional Bézier curves by adopting Non-Uniform Rational B-Splines (NURBS), which allow for localized adjustments along the curve and better adaptation to complex loss landscapes.
3. Plug-and-play compatibility: EXTRA can be seamlessly integrated with existing multi-task learning methods (e.g., Linear Scalarization, FairGrad, CAGrad), demonstrating strong modularity and practical applicability across different optimization strategies.

Weaknesses
1. The use of a single continuous curve assumes a smooth task relationship in parameter space, which may not hold in complex real-world MTL settings with nonlinear or conflicting task interactions. This is related with EXTRA is primarily designed for two-task scenarios by learning a curve between two endpoints. Extending to settings with three or more tasks or modeling high-dimensional Pareto fronts (e.g., via surfaces) remains challenging. The paper avoid this by task grouping, I'm not sure it is enough.
2. The method requires separate training of single-task endpoints followed by curve optimization, introducing additional computational cost and implementation complexity compared to end-to-end approaches.
3. The effectiveness of the proposed grouping strategy relies on gradient similarity computed during the warm-up stage. Inaccurate grouping due to noisy gradients or ill-defined task similarity may negatively affect performance.
4. The paper omits evaluation against recent state-of-the-art optimization-based MTL methods such as Go4align [1] and Selective Task Group Updates [2], which provide finer-grained control over task gradients and group-level optimization. Including such baselines would better position EXTRA in the evolving MTL landscape.

[1] Go4align: Group optimization for multi-task alignment
[2] Selective Task Group Updates for Multi-Task Optimization

5. Despite introducing a novel empirical approach, the paper lacks formal theoretical guarantees. There is no analysis of convergence, approximation quality to the Pareto front, or how the order-aware loss formally shapes the learned curve. This leaves the method’s success purely justified by empirical performance, without insight into when or why it should be expected to succeed.

---

> ### Author Rebuttal · Authors · 2025-07-30
>
> We thank the Reviewer 5udz's constructive and insightful comments. To address your questions, we provide pointwise responses below.
>
> >* Q1: The use of a single continuous curve assumes a smooth task relationship in parameter space, which may not hold in complex real-world MTL settings with nonlinear or conflicting task interactions. This is related with EXTRA is primarily designed for two-task scenarios by learning a curve between two endpoints. Extending to settings with three or more tasks or modeling high-dimensional Pareto fronts (e.g., via surfaces) remains challenging. The paper avoid this by task grouping, I'm not sure it is enough.
>
>   A1: We appreciate the reviewers' concerns but believe some misunderstandings have arisen. We address these concerns point by point: (1) Achieving a completely smooth task relationship on a single curve is indeed challenging. As shown in Figure 4 of the main text, not all points on the trained curve lie within the low-loss region. Without the curve generated by our method, a satisfactory trade-off cannot be obtained. This is why we employ an order-aware objective and NURBS. The order-aware objective encourages exploration of multi-task learning (MTL) trade-off points, which may be widely separated, while the flexibility of NURBS compensates for this. It is important to clarify that we do not expect the trained curve to fully realize the MTL trade-off; rather, we aim to optimize the curve to approximate these trade-off points as closely as possible. Experimental results demonstrate the effectiveness of this approach for MTL. (2) Extending this paradigm to settings with many tasks is indeed challenging. However, our insights are consistent with the two referenced papers [1, 2], both of which group tasks based on inherent correlations and achieve balanced learning across groups.
>
>
>
> >* Q2: The method requires separate training of single-task endpoints followed by curve optimization, introducing additional computational cost and implementation complexity compared to end-to-end approaches.
>
>   A2: We understand the reviewer’s concern and acknowledge that our method introduces additional computational overhead due to its multi-stage training paradigm. However, we would like to emphasize that this overhead is affordable and justified by the significant benefits it brings.
>
>   First, our approach offers a novel research perspective for MTL, departing from conventional gradient manipulation techniques whose effectiveness has recently been questioned by emerging literature [3,4]. Second, the proposed paradigm enables explicit exploration of task trade-offs, which leads to significant improved performance. Finally, our method is compatible with mainstream MTL approaches, as demonstrated by our plug-and-play experiments, underscoring its flexibility and broad applicability.
>
>
>
> >* Q3: The effectiveness of the proposed grouping strategy relies on gradient similarity computed during the warm-up stage. Inaccurate grouping due to noisy gradients or ill-defined task similarity may negatively affect performance.
>
>   A3: Thank you for the valuable question. We appreciate the reviewers’ concerns. In this work, we simply propose a grouping strategy to demonstrate the effectiveness of our framework. As shown in Figure 7 (Appendix), we compare multiple grouping methods, and the results indicate that even random task division can yield competitive performance. Compared to the random strategy, our method groups more similar tasks together, which leads to improved outcomes.
>
>
>
> >* Q4: The paper omits evaluation against recent state-of-the-art optimization-based MTL methods such as Go4align [1] and Selective Task Group Updates [2], which provide finer-grained control over task gradients and group-level optimization. Including such baselines would better position EXTRA in the evolving MTL landscape.
>
>   A4: Thank you for the valuable question. We agree that it is important to include these methods in the comparison. Below, we present the results (* indicates values reported in their original paper). It is worth noting that SelectiveMTL appears to perform poorly on CityScapes and NYUv2. This is likely because the original SelectiveMTL paper evaluates on different datasets than ours. Therefore, we re-implemented the method based on their official code. However, this comparison may not be entirely fair. Due to time constraints, we were unable to conduct thorough hyperparameter tuning, but we plan to include a more carefully tuned comparison with SelectiveMTL in a future version of this work.
>
> |              | CityScapes | NYUv2 |
> | ------------ | ---------- | ----- |
> | SelectiveMTL | 15.28      | -0.22 |
> | GO4Align*    | 8.11       | -6.08 |
> | EXTRA-F      | 1.63       | -6.30 |
>
>
>
> >* Q5: Despite introducing a novel empirical approach, the paper lacks formal theoretical guarantees. There is no analysis of convergence, approximation quality to the Pareto front, or how the order-aware loss formally shapes the learned curve. This leaves the method’s success purely justified by empirical performance, without insight into when or why it should be expected to succeed.
>
>   A5: We acknowledge that this paper does not provide a formal theoretical analysis of our approach. This is primarily due to the complex loss landscape and the fundamentally different learning paradigm compared to gradient manipulation-based methods. Currently, there is no existing research that offers theoretical guarantees for the convergence of mode connectivity.
>
>   Nonetheless, we would like to offer some intuition in this regard. Consider a two-task scenario as an example. At $t = 0$, we have $\nabla \mathcal{L}_1(\mathcal{C}(0;\theta)) = 0$, and at $t = 1$, $\nabla \mathcal{L}_2(\mathcal{C}(1;\theta)) = 0$. This property arises from using pre-trained single-task weights as the endpoints. Suppose we reasonably assume that these two endpoints are the closest local minima to each other, which is supported by empirical evidence suggesting that mode connectivity works well when endpoints are nearby [5]. This also aligns with the intuition in MTL that tasks should be correlated. If this assumption holds, then it follows that $\nabla \mathcal{L}_2(\mathcal{C}(0;\theta)) \cdot \nabla \mathcal{L}_1(\mathcal{C}(1;\theta)) < 0$.
>
>   Now define $F(t) = \nabla \mathcal{L}_1(\mathcal{C}(t;\theta)) + \nabla \mathcal{L}_2(\mathcal{C}(t;\theta))$. Due to the $C^1$ continuity of NURBS and the differentiability of $\mathcal{L}_1$ and $\mathcal{L}_2$, $F(t)$ is continuous on the interval $[0, 1]$. By the Bolzano–Cauchy Theorem, there exists $t' \in (0,1)$ such that
>   $
>   F(t') = \nabla \mathcal{L}_1(\mathcal{C}(t';\theta)) + \nabla \mathcal{L}_2(\mathcal{C}(t';\theta)) = 0,
>   $
>   which implies the existence of a Pareto stationary point.
>
>   We emphasize that this is not a rigorous theoretical analysis, but rather a conceptual insight. Nevertheless, our contribution lies in introducing a novel perspective to MTL, distinct from gradient manipulation-based approaches. This is particularly meaningful given the ongoing debate over the practical effectiveness of gradient-based MTL methods [3,4]. Furthermore, we properly adapt and apply mode connectivity to MTL, differing significantly from its prior use in PFL.
>
>
>
>
>
>   **Reference**:
>
>   [1] Go4align: Group optimization for multi-task alignment. NeurIPS 2024.
>
>   [2] Selective Task Group Updates for Multi-Task Optimization. ICLR 2025.
>
>   [3] Robust Analysis of Multi-Task Learning Efficiency: New Benchmarks on Light-Weighed Backbones and Effective Measurement of Multi-Task Learning Challenges by Feature Disentanglement, arXiv 2024;
>
>   [4] Can Optimization Trajectories Explain Multi-Task Transfer?, arXiv 2024;
>
>   [5] Optimizing Mode Connectivity via Neuron Alignment. NeurIPS 2020.

---

> > ### Comment · Reviewer_5udz · 2025-08-05
> >
> > I have reviewed the authors’ full response. While the lack of concrete convergence analysis remains a weakness—given its essential role in most prior work—I believe the paper is more deserving of acceptance than rejection if my concerns are adequately addressed in a future revision. Accordingly, I have adjusted my ratings

---

> > > ### Author Response · Authors · 2025-08-05
> > >
> > > Thank you for your reply. We are pleased to have addressed most of your concerns and will make sure to incorporate these responses and improvements into our revision.

---

### Official Review · Reviewer_XCec · 2025-06-29

**Clarity:** 2
**Significance:** 3
**Originality:** 3
**Rating:** 4
**Confidence:** 4

**Summary:**

The authors propose a novel curve-based optimization for locating the MTL tradeoffs, in contrast to previous end-point-based approaches. Discover and tackle 3 challenges of curve-based mode connectivity. Extensive experiments show superior performance of the proposed methods compared to various baselines, including single-point MTL as well as pareto front learning.

**Questions:**

1) In Figure 4 (a)-(d), it is not very clear what the x- and y-axes represent in the loss landscape plots.

2) In Line 217, the authors state “Figure 4(c)(g) provides a preview of the benefits of our approach”, where (c)(g) correspond to the order-aware Bézier curve. It is confusing if the order-aware Bézier curve is the authors' approach or if it is the NURBS curve (corresponding to (d)(h)).

3) In Equation 2, there is no assumption on the relative magnitude of losses at t1 and t2. Is this made on-purpose? If the loss at t1 is made smaller than that at t2 significantly the objective is still minimized.

4) The purpose of the additional objective defined in Equation 2 seems still unclear. By minimizing the objective, it seems that we wish the losses at two sampled points t1 and t2 be similar in magnitude?

5) In Equation 2, why does the first term contain (t2-t1) while the second term has a reversed (t1-t2), that is t1 comes first? What is the underlying principle and how does it extend to three or more tasks?

6) How does the method work in a two-task scenario, where it is impossible to make three clusters?

7) In ablation (e.g. Section B.3), does EXTRA correspond to EXTRA-L or EXTRA-F? This is important in terms of training efficiency as to my knowledge and experience FairGrad requires significantly larger training time due to per-task gradient sampling as well as its quadratic optimization.

8) In Line 591, the authors state “we observe that two of the three K-means clusters contain only a single task”. If a cluster contains only a single task, how does MTL training method such as FairGrad apply? As it degenerates to a STL on this particular cluster. I suppose no MTL is done in such a case?

**Ethical Concerns:**

["NO or VERY MINOR ethics concerns only"]

**Final Justification:**

The authors carefully responded to each of my questions and concerns, and indeed cleared most of my doubts. The experiments are extended, which convinces the reader that the proposed method indeed works great. The revision is needed to include extended results and clarify points referred to in our discussion.

**Limitations:**

Yes.

**Quality:**

3

**Strengths And Weaknesses:**

### Strengths
1) Proposing a curve-based mode connectivity approach tailored for MTL is novel, as vanilla methods (such as Bézier curves) have significant limitations. This brings a new view to the MTL field, which I personally think is valuable.

2) Identifies three key challenges of curve-based mode connectivity for MTL, with experimental evidence, bringing insights to the community. Presents three designs tailored to resolve the three challenges, demonstrating a systematic and well-motivated algorithmic design.

3) Extensive experiments show superior performance of the proposed algorithm, setting a new SoTA for MTL.


### Major Weaknesses
1) The authors state that the additional objective (Equation 2) is introduced to resolve challenge 1. While from ablations we see the importance of this additional objective for performance, it is unclear if this objective really tackles challenge 1. Specifically, a plot similar to Figure 4, which is used to illustrate challenge 1, is desirable, with and without the additional objective. Anyhow the claim that Equation 2 resolves challenge 1 requires careful analysis to be backed.

2) The motivational examples (Figure 4) to illustrate challenge 1 and 2 should be conducted on more than one dataset to show the generality of the observation. This is crucial if a scientific claim is to be made.

3) The effect of the number of control points shall be studied more carefully on more settings (such as another dataset or another model). It is unclear if the number 4 is an in general best setting and the phenomenon of irregular curves with higher number of control points hold for other settings.


### Minor Weaknesses
1) Line 97, missing space after SPRO, and some others such as in Line 107. This is inconsistent with e.g. Line 98, CBFT, I recommend being consistent.

2) Line 146, “we introduce the our application …” is a grammar error.

3) Baselines such as MOCO, AlignedMTL, STCH, are all recent representative MTL algorithms, what is the particular reason that they are not included in the comparison to facilitate evaluation that is more up-to-date?

4) The axis labels in Figure 4 and 9 are very small, making them a bit hard to read.

5) In Line 578, the symbol “L_ast” is never stated, and I am not sure what it corresponds to. Do you mean L_o? Also L_align and L_o in this section is inconsistent with R_align and R_o in the main paper.

---

> ### Author Rebuttal · Authors · 2025-07-30
>
> We thank the Reviewer XCec's constructive and insightful comments. To address your questions, we provide pointwise responses below.
>
> >* Q1: The authors state that the additional objective (Equation 2) is introduced to resolve challenge 1. While from ablations we see the importance of this additional objective for performance, it is unclear if this objective really tackles challenge 1. Specifically, a plot similar to Figure 4, which is used to illustrate challenge 1, is desirable, with and without the additional objective. Anyhow the claim that Equation 2 resolves challenge 1 requires careful analysis to be backed.
>
> A1: As shown in Figures 4(f) and 4(g), the points along the curve generally demonstrate strong MTL performance, which can be attributed to mode connectivity. However, due to randomness (Challenge 1), Figure 4(f) also exhibits fluctuations in the trade-off performance (reflected in $\Delta m%$). To address this, our order-aware objective $R_o$ enables the trade-off to be more evenly traversed along the curve, thereby mitigating this challenge. An ablation study of $R_o$, presented in Table 4 (Appendix), further highlights its contribution to our system. We apologize for the mislabeling of $R_o$ as $L_o$ in Table 4. Lastly, we would like to clarify that Figure 4 presents experimental results on the Cityscapes dataset, which we believe holds strong practical relevance.
>
> >* Q2: The motivational examples (Figure 4) to illustrate challenge 1 and 2 should be conducted on more than one dataset to show the generality of the observation. This is crucial if a scientific claim is to be made.
>
> A2:  Due to policy constraints, we are unable to upload these figures at this time. Therefore, we list and compare the MTL performance on **NYUv2** between **vanilla Bézier** and **Bézier + order-aware**, and present their sampled results below. As shown, a similar trend is observed to that on the CityScapes dataset.
>
>   | t                    | 0.1   | 0.3   | 0.5   | 0.7   | 0.9   |
>   | -------------------- | ----- | ----- | ----- | ----- | ----- |
>   | Vanilla Bezier       | -1.22 | -0.67 | -3.60 | -4.00 | -3.42 |
>   | Bezier + order-aware | -1.01 | -3.52 | -4.36 | -3.81 | -3.35 |
>
> >* Q3: The effect of the number of control points shall be studied more carefully on more settings (such as another dataset or another model). It is unclear if the number 4 is an in general best setting and the phenomenon of irregular curves with higher number of control points hold for other settings.
>
> A3: Thank you for your valuable question. In Section A.2 of the Appendix, we provide the control point settings for each dataset: 4 for Cityscapes, 5 for NYUv2, and 3 for both CelebA and MultiMNIST. To further address your concern, we conducted additional experiments on NYUv2 with varying numbers of control points, and present the results below. Overall, all variants outperform the baseline, showing a consistent trend with Cityscapes (Figure 9 in the Appendix): using fewer control points reduces flexibility, while using too many introduces optimization difficulties. Both extremes lead to suboptimal performance.
>
>   | Control Number | $\Delta m %$ |
>   | -------------- | ---------- |
>   | 4              | -6.15      |
>   | 5              | -6.30      |
>   | 6              | -5.57      |
>
>
>
> >* Q4: Why don't include more update-to-date approaches, e.g., MoCo, AlignedMTL, and STCH, etc.?
>
> A4: Thank you for your valuable suggestion. We have included these methods in our comparison, with the results presented below (* indicates values reported in the original papers). As shown, EXTRA continues to demonstrate superior performance. It is important to note that the performance of STCH on the CityScapes dataset is based on our re-implementation, as the original paper did not report results for this dataset. Therefore, this result may not accurately reflect its true performance. Due to time constraints, we were unable to perform thorough hyperparameter tuning. However, the result for STCH on NYUv2 is taken directly from their original paper.
>
>   |           | CityScapes | NYUv2 |
>   | --------- | ---------- | ----- |
>   | MoCo*     | 9.90       | 0.16  |
>   | AlignMTL* | -          | -4.93 |
>   | STCH      | 12.37      | -1.35 |
>   | EXTRA-F   | 1.63       | -6.30 |
>
>
>
> >* Q5: In Line 578, the symbol $L_{ast}$ is never stated, and I am not sure what it corresponds to. Do you mean $L_o$? Also $L_{\text{align}}$ and $L_o$ in this section is inconsistent with $R_{\text{align}}$ and $R_o$ in the main paper.
>
> A5: We apologize for the typos. In this paragraph, $L_o$ and $L_{\text{align}}$ should be $R_o$ and $R_{\text{align}}$, respectively. Additionally, $L_{\text{ast}}$ refers to the same objective as $R_{\text{align}}$.
>
>
>
> >* Q6: In Figure 4 (a)-(d), it is not very clear what the x- and y-axes represent in the loss landscape plots.
>
> A6: Our visualization of implementation is adapted from [1]. Suppose we have three weight vectors (control points) $w_1$, $w_2$, $w_3$. We set $u = (w_2 - w_1)$, $v = (w_3 - w_1) - \langle w_3 - w_1, w_2 - w_1 \rangle / \\| w_2 - w_1 \\|^2 \cdot (w_2 - w_1)$. Then the normalized vectors $\hat{u} = u / \\| u \\|$, $\hat{v} = v / \\| v \\|$ form an orthonormal basis in the plane containing $w_1, w_2, w_3$. To visualize the loss in this plane, we define a Cartesian grid in the basis $\hat{u}, \hat{v}$ and evaluate the networks corresponding to each of the points in the grid. A point P with coordinates $(x, y)$ in the plane would then be given by $P = w_1 + x \cdot \hat{u} + y \cdot \hat{v}$.
> We will explicitly involve these explaination in our next version.
>
>
>
> >* Q7: In Line 217, the authors state “Figure 4(c)(g) provides a preview of the benefits of our approach”, where (c)(g) correspond to the order-aware Bézier curve. It is confusing if the order-aware Bézier curve is the authors' approach or if it is the NURBS curve (corresponding to (d)(h)).
>
> A7: Figure 4(c) and 4(g) illustrate the effect of incorporating the order loss into the Bézier curve, highlighting its benefits for trade-off exploration. We will revise the text to explain this point more explicitly.
>
>
>
> >* Q8: In Equation 2, there is no assumption on the relative magnitude of losses at t1 and t2. Is this made on-purpose? If the loss at t1 is made smaller than that at t2 significantly the objective is still minimized.
>
> A8: Currently, we have not explicitly addressed the issue of loss magnitude. However, we fully understand the reviewer's concern. Prior studies have indeed highlighted the issue of scale differences between task losses, which can cause imbalance during the optimization of $R_o$. Nonetheless, based on our evaluation results, the current formulation already achieves the intended trade-off effect. We appreciate the reviewer’s valuable suggestion and will consider investigating this issue in future work.
>
>
>
> >* Q9: The purpose of the additional objective defined in Equation 2 seems still unclear. By minimizing the objective, it seems that we wish the losses at two sampled points t1 and t2 be similar in magnitude?
>
> A9: We would like to clarify that this objective is designed to endow the learned curve with trade-off property between tasks similar to the that owns by Pareto front. During training, by optimizing this objective, we can not only maintain the low loss of overall task loss, but also encourage the traverse trade-offs, thereby assisting explore the best trade-off. Specifically, we achieve this by encouraging $L_1$ to be monotonically increasing (i.e., $L_1(t_1) < L_1(t_2)$, if $t_2 > t_1$) and $L_2$ monotonically decreasing (i.e., $L_2(t_1) > L_2(t_2)$, if $t_2 > t_1$) when t ranges from [0, 1].
>
>
>
> >* Q10: How does Equation 2 extend to three or more tasks? How does the method work in a two-task scenario, where it is impossible to make three clusters?
>
> A10:  We would like to clarify that we employ Eqn. (3) to tackle two-task scenarios, while employ Eqn. (7) (with additional designed $R_{\text{align}}$ and grouping strategy if encounter massive tasks) to tackle many-task scenarios.
>
>
>
> >* Q11: In ablation (e.g. Section B.3), does EXTRA correspond to EXTRA-L or EXTRA-F? This is important in terms of training efficiency as to my knowledge and experience FairGrad requires significantly larger training time due to per-task gradient sampling as well as its quadratic optimization.
>
> A11:  The EXTRA implementation discussed in Section B.3 specifically refers to EXTRA-F, which is included to highlight its memory-efficient property in comparison to endpoint-based mode connectivity approaches. We acknowledge the reviewer's concern regarding the potential inefficiency of EXTRA-F in terms of training time, given that it adopts FairGrad as its baseline. This is a valid concern. However, we would like to provide further clarification from two perspectives: (1) EXTRA-L already achieves SOTA performance, demonstrating the strength of our proposed method. (2) The EXTRA framework is orthogonal to existing MTL approaches. Even when combined with strong SOTA baselines, it consistently yields significant improvements, as confirmed by our plug-and-play experiments presented in Section B.4 (Appendix).
>
>
>
> >* Q12: In Line 591, the authors state “we observe that two of the three K-means clusters contain only a single task”. If a cluster contains only a single task, how does MTL training method such as FairGrad apply? As it degenerates to a STL on this particular cluster. I suppose no MTL is done in such a case?
>
> A12: Under this scenario, clusters containing only a single task do not participate in the first training stage. Instead, they are solely used to compute the overall task loss $L_T$ and the alignment regularization term $R_{\text{align}}$ in Equation (7).
>
>
> >* Q13: Typos or unclear figures.
>
> A13: Many thanks to your reminder, we will fix them and proofread the paper.
>
>   **Reference**:
>
>   [1] Loss Surfaces, Mode Connectivity, and Fast Ensembling of DNNs. NeurIPS 2018.

---

> > ### Comment · Reviewer_XCec · 2025-08-04
> >
> > Thanks for providing a detailed response.
> >
> > I’ll present point-wise comment to the authors’ response:
> >
> > **Response 1**
> >
> > Indeed, comparing Figure 4 (d) and (g), we see that injecting the order-aware objective on Bezier curve learning gives less fluctuation, this supports the claim that the introduced objective is useful in resolving Challenge 1. I apologize for missing this out.
> >
> > However, I would still be interested in seeing if with and without this objective, in your final algorithm, would result in one with fluctuation and the other that is more stable. Note that your ablation in Table 4 does not reflect this observation because you only present a single number (a single deltaM) for each ablation. I would leave this as something that I hope you include in your updated manuscript but which does not affect my current recommendation decision.
> >
> > **Response 2**
> >
> > Nice, from this we can also see that with your order-aware objective, the Bezier curve gives less fluctuation in terms of deltaM on NYUv2. This is consistent with the observation on CityScapes. I would recommend adding proper plots in the updated manuscript (with denser evaluation of t, like in your Figure 4).
> >
> > **Response 3**
> >
> > I appreciate the added experiment, and the result is convincing as a number around 4-5 is in general a good default setting. I would recommend adding a plot similar to Figure 9 in your Appendix for NYUv2, to show under 6 control points, what the curve looks like, is it irregular?
> >
> > **Response 4, 6**
> >
> > I appreciate the added experiment; now the result seems more complete and up-to-date.
> >
> > I appreciate the clarification.
> >
> > **Response 7**
> >
> > Yes, a clarification on what exact part of your contribution is previewed in Figure 4 would largely improve the readability of the paper.
> >
> >
> > **Response 8**
> >
> > Yes, despite the empirically good performance, I would still be interested in a deeper analysis of this objective, hope you could dig deeper in the future.
> >
> >
> > **Response 9**
> >
> > Now this makes the intuition behind your objective very clear. I would highly recommend adding such an explanation in your manuscript to facilitate better understanding of your contribution (I apologize if you already included such an explanation but I miss it).
> >
> >
> > **Response 10**
> >
> > Now I am clear with the fact that Equation 2 is applied for two-tasks and Equation 7 is applied when K>2 (with task grouping). However, I am still a bit confused since Equation 7 includes Equation 2 (by R_o). Does this mean you apply Equation 7 to each pair of tasks in cluster 3? I think I still miss some points here. How is R_o inside Equation 7 defined if Equation 7 is applied to more than two tasks?
> >
> >
> > **Response 5, 11, 12, 13**
> >
> > Noted.

---

> > > ### Author Response · Authors · 2025-08-04
> > >
> > > Thank you for your prompt response. We are pleased to have addressed most of your concerns, and we greatly appreciate your constructive feedback. We will incorporate your suggestions in the updated version, including clearer explanations, additional visualizations, and more experimental results. Below, we respond to your remaining concerns:
> > >
> > > > * Q1: However, I would still be interested in seeing if with and without this objective, in your final algorithm, would result in one with fluctuation and the other that is more stable. Note that your ablation in Table 4 does not reflect this observation because you only present a single number (a single deltaM) for each ablation. I would leave this as something that I hope you include in your updated manuscript but which does not affect my current recommendation decision.
> > >
> > > A1: Your concern may require evaluating a large number of random seeds, which may not be feasible at this time. Nevertheless, we will try to expand our observations and include the results in the updated version of the manuscript. In our current experiments, we have not observed such fluctuations. Intuitively, even if $R_o$ is not minimized effectively, some fluctuations may remain, but they are still mitigated compared to the setting without $R_o$ constraints. To demonstrate the robustness of $R_o$, we report the specific performance across three random seeds under both conditions—with and without $R_o$—in the hope of addressing your concern.
> > >
> > > |            | 42    | 43    | 44    |
> > > | ---------- | ----- | ----- | ----- |
> > > | with $R_o$ | -5.29 | -5.92 | -7.69 |
> > > | w/o $R_o$  | -4.72 | -5.75 | -4.45 |
> > >
> > > > * Q2: I would recommend adding a plot similar to Figure 9 in your Appendix for NYUv2, to show under 6 control points, what the curve looks like, is it irregular?
> > >
> > > A2: Thank you for the helpful suggestion. Due to current policy restrictions, we are unable to upload additional figures at this time. While numerical values alone cannot convey the insights, we will incorporate the requested visualization and corresponding analysis in the appendix of the updated version.
> > >
> > > > * Q3: However, I am still a bit confused since Equation 7 includes Equation 2 (by $R_o$). Does this mean you apply Equation 7 to each pair of tasks in cluster 3? I think I still miss some points here. How is $R_o$ inside Equation 7 defined if Equation 7 is applied to more than two tasks?
> > >
> > > A3: We appreciate your question and would like to clarify. In our implementation of Equation 7:
> > >
> > > - Tasks are first grouped into three clusters.
> > > - We apply the $R_o$ objective to **two** of these clusters to model trade-offs.
> > > - The remaining **third cluster** is optimized using $R_{align}$, which encourages alignment of task losses around $t=0.5$.
> > >
> > > This design allows us to explore trade-offs between two clusters while simultaneously guiding the third cluster to align toward the midpoint. This strategy is consistent with our broader experimental protocol.

---

> > > > ### Comment · Reviewer_XCec · 2025-08-05
> > > >
> > > > Thank the authors for addressing my questions. I will maintain the current positive rating and hope authors can update their paper accordingly afterward; the final rating may be updated again based on the AC/reviewer discussion panel.

---

> > > > > ### Author Response · Authors · 2025-08-05
> > > > >
> > > > > We thank the reviewer for the positive feedback. We will ensure that the responses and suggested improvements are carefully incorporated into our revised version.

---

### Official Review · Reviewer_KFeN · 2025-07-02

**Clarity:** 3
**Significance:** 3
**Originality:** 4
**Rating:** 5
**Confidence:** 4

**Summary:**

This paper presents a novel multi-task learning (MTL) framework based on mode connectivity, where the goal is to optimise the curve that connects task-specific endpoints (i.e., task-specific weights), rather than optimising the endpoints themselves as in conventional MTL methods. This leads to a curve-based optimisation strategy for Pareto front approximation, offering an alternative to standard gradient- and weighting-based MTL methods. The authors also provide a thoughtful discussion of the challenges specific to curve-based Pareto optimisation, along with design improvements to mitigate them. The proposed method is evaluated on multiple multi-task benchmarks across scene understanding and image classification tasks.

**Questions:**

See weaknesses.

**Ethical Concerns:**

["NO or VERY MINOR ethics concerns only"]

**Final Justification:**

The authors have sufficiently resolved my concerns, and therefore I am happy to keep the original rating.

**Limitations:**

No. would be great if the authors could discuss some remaining challenges that need to be solved.

**Paper Formatting Concerns:**

No.

**Quality:**

3

**Strengths And Weaknesses:**

The paper is well written and clearly structured. The proposed approach is both unique and insightful, and the connections drawn to existing MTL paradigms are well articulated. Visualisations effectively support different design insights, particularly in how the design addresses different optimisation challenges.

[Weaknesses and Discussion Points]
- Given the stochastic nature of curve-based optimisation, how sensitive is the final learned control point to different choices of hyperparameters, network architectures, or initialisations? Some empirical evidence or discussion on training stability would be helpful.
- Can we extract any structural insights from the learned curves themselves? For example, if the final Pareto curve exhibits low curvature (i.e., nearly linear), might that imply the tasks are closely aligned or share similar representations? Conversely, does high curvature suggest more divergent task optima?
- Some MTL baselines, particularly those using second-order gradients or iterative local optimisation (e.g., Auto-Lambda, PCGrad variants), can be computationally expensive. How does the runtime of the proposed method compare to such baselines? An analysis of training time or convergence speed would provide a more complete evaluation.
- Figure 6 is difficult to interpret and contributes little beyond what simpler metrics already show. Since Multi-MNIST is a relatively easy dataset with small performance differences, it may not be the best case for visualising Pareto cruves. I recommend including similar visualisations for more complex datasets, where task trade-offs are more prominent and visual differences are more informative.

---

> ### Author Rebuttal · Authors · 2025-07-30
>
> We thank the Reviewer KFeN's constructive and insightful comments, and recognize our contributions. To address your questions, we provide pointwise responses below.
>
> > * Q1: Given the stochastic nature of curve-based optimisation, how sensitive is the final learned control point to different choices of hyperparameters, network architectures, or initialisations? Some empirical evidence or discussion on training stability would be helpful.
>
> A1: Thank you for your valuable question. We have evaluated our method across various tasks (e.g., image classification, scene understanding), architectures (e.g., CNN, MTAN), and hyperparameters (see Figures 9–11). To further address the reviewer's concern, we have conducted additional experiments focusing on initialization strategies. Specifically, the initialization of the curve is determined by the initialization of control points. In our current design, the control points are randomly initialized. For comparison, we consider a linear initialization scheme, where control points are initialized via linear interpolation between two pre-trained endpoints. The results on the Cityscapes dataset are presented below.
>
> As observed, although linear initialization yields slightly worse performance compared to random initialization, it still achieves SOTA MTL performance. This may be attributed to the complex nature of NURBS curves, which differ significantly from linear curves, suggesting that linear initialization might not be optimal for our framework.
>
>   |                        | mIoU  | Pix. Acc. | Abs. Err. | Rel. Err. | $\Delta$ m % |
>   | ---------------------- | ----- | --------- | --------- | --------- | ---------- |
>   | Linear initialisations | 73.91 | 93.42     | 0.0129    | 29.40     | 2.25       |
>   | Random initialisations | 76.11 | 93.58     | 0.0126    | 30.20     | 1.63       |
>
> > * Q2: Can we extract any structural insights from the learned curves themselves? For example, if the final Pareto curve exhibits low curvature (i.e., nearly linear), might that imply the tasks are closely aligned or share similar representations? Conversely, does high curvature suggest more divergent task optima?
>
> A2:  Thank you for your valuable question. Intuitively, low curvature may indicate a high degree of correlation between tasks, whereas high curvature suggests lower task correlation. Although, to the best of our knowledge, no existing work directly establishes this connection, some studies offer relevant insights. For instance, [1] demonstrates that if two models (i.e., local minima) learn similar functions, a low-loss connecting path between them is more likely to exist. Nevertheless, this observation is not yet conclusive, and we plan to explore this relationship more thoroughly in future work.
>
>
>
> > * Q3: Some MTL baselines, particularly those using second-order gradients or iterative local optimisation (e.g., Auto-Lambda, PCGrad variants), can be computationally expensive. How does the runtime of the proposed method compare to such baselines? An analysis of training time or convergence speed would provide a more complete evaluation.
>
> A3: Thank you for your valuable question. As suggested, we compare the training cost of our method, **EXTRA-L**, with **PCGrad**, and present the results below. All training times are measured using a single A100 GPU. As observed, although EXTRA-L incurs additional cost compared to its baseline (LS) due to the multi-stage training paradigm, it remains slightly more efficient than PCGrad.
>
>   |         | Time COST (minus) |
>   | ------- | ----------------- |
>   | LS      | 140.2             |
>   | PCGrad  | 508.8              |
>   | **EXTRA-L** | **475.6**             |
>
>
>
> > * Q4: Figure 6 is difficult to interpret and contributes little beyond what simpler metrics already show. Since Multi-MNIST is a relatively easy dataset with small performance differences, it may not be the best case for visualising Pareto cruves. I recommend including similar visualisations for more complex datasets, where task trade-offs are more prominent and visual differences are more informative.
>
> A4: Thank you for your valuable suggestion. Due to policy constraints, we are unable to upload any figures at this stage. Therefore, we provide the task-wise performance at each preference in tabular form instead. According to the listed results, all methods achieve certain trade-offs, but EXTRA obtains the best overall performance due to its effective exploration of the trade-off space.
>
> Currently, the widely used two-task PFL datasets (e.g., MNIST variants, Census, etc.) are all small in scale and thus face the same limitation you pointed out. We will explore more complex datasets as alternatives for a more meaningful evaluation of Pareto fronts in future work.
>
>   | EXTRA  | 0.912, 0.960 | 0.937, 0.957 | 0.940, 0.952 | 0.958,0.949  | 0.963, 0.925 |
>   | ------ | ------------ | ------------ | ------------ | ------------ | ------------ |
>   | PaMaL  | 0.927, 0.933 | 0.932, 0.936 | 0.940, 0.939 | 0.945, 0.930 | 0.944, 0.915 |
>   | COSMOS | 0.825, 0.912 | 0.898, 0.908 | 0.910, 0.900 | 0.921, 0.890 | 0.923, 0.770 |
>
>
>
> > * Q5: Limitation: No. Would be great if the authors could discuss some remaining challenges that need to be solved.
>
>   A5: We would like to respectfully point out that the discussion of our method's limitations is explicitly provided in Section C of the appendix accompanying the main submission. We chose to include it in the appendix to maintain a more focused presentation of experimental results in the main text. In future revisions, we will consider moving this discussion into the main body for greater visibility.
>
> **Reference**:
>
> [1] Git Re-Basin: Merging Models modulo Permutation Symmetries. ICLR 2023.

---

> > ### Comment · Area_Chair_SXYM · 2025-08-05
> >
> > Dear Reviewer KFeN
> >
> > The authors have updated their detailed responses to your concerns. As the Author-Reviewer discussion period ends, please confirm whether your concerns were addressed and whether this may inform any changes to your score.
> >
> > Best regards,
> > AC

---

> ### Author Response · Authors · 2025-08-08
>
> Dear Reviewer KFeN,
>
> As the deadline for the author–reviewer discussion phase approaches, we would like to kindly ask whether our responses have addressed your concerns.
>
> Best regards,
>
> Authors

---

> > ### Comment · Reviewer_KFeN · 2025-08-08
> > **Final comment**
> >
> > Hi,
> >
> > Thanks for your detailed rebuttal. I have no further questions, and happy to keep my original rating.

---

> > > ### Author Response · Authors · 2025-08-08
> > >
> > > Thank you for your positive feedback!

---

### Official Review · Reviewer_uJP2 · 2025-07-05

**Clarity:** 2
**Significance:** 2
**Originality:** 2
**Rating:** 4
**Confidence:** 2

**Summary:**

This paper proposes EXTRA (EXplore TRAde-offs) for multi-task learning (MTL) that leverages mode connectivity to find optimal trade-offs between multiple task objectives. Unlike most optimization-based MTL methods that adjust the gradient updates for shared parameters, EXTRA models the parameter space as a curve connecting solutions for individual tasks, efficiently exploring the Pareto front of multi-objective trade-offs.


Specifically, it introduces Non-Uniform Rational B-Splines (NURBS) for flexible curve modeling in the parameter space, improving on previous methods like Bézier curves.
It proposes an order-aware objective to structure the search along the curve, promoting better trade-offs. The method achieves SOTA performance across several MTL benchmarks (CityScapes, NYUv2, CelebA, MultiMNIST).

**Questions:**

1. How well would this approach generalize to other tasks/objectives, model architectures with different loss landscape properties?

2. How does the method compare with existing ones in terms of computational complexity?

**Ethical Concerns:**

["NO or VERY MINOR ethics concerns only"]

**Final Justification:**

Most of my concerns are addressed in the rebuttal.

**Limitations:**

- Limited theoretical guarantees on Pareto optimality or convergence.

**Paper Formatting Concerns:**

No concerns.

**Quality:**

3

**Strengths And Weaknesses:**

### Strengths

1. Extensive experiments on MTL benchmarks are performed to show the effectiveness of the proposed method.

2. The paper is clearly written.

3. The proposed method seems to provide a new perspective to explicitly consider the landscape of the multi-objective optimization problem.


### Weaknesses


1. The proposed method with curve optimization seems to add computational overhead. See Questions-2.


2. Some of the motivation for the methodology is not clearly discussed. It is unclear why the objective in Section 5.1 is designed in this way. More explanation should be given.


3. Mode connectivity and curve optimization is not new. The paper seems to combine and apply existing methods to multi-task learning. It would be better to state the difference of these two in multi-task compared to single-task learning.


4. There are some important missing references in multi-task learning/multi-objective optimization and mode connectivity which the authors should discuss. I list a few below as examples.

4.1 The following papers on multi-objective optimization should be discussed.

- Steepest descent methods for multicriteria optimization

- Three-Way Trade-Off in Multi-Objective Learning: Optimization, Generalization and Conflict-Avoidance


4.2 The following papers on Pareto front approximation with MultiMNIST experiments should be discussed and compared.

- Pareto Multi-Task Learning

- Multi-Task Learning with User Preferences: Gradient Descent with Controlled Ascent in Pareto Optimization

---

> ### Author Rebuttal · Authors · 2025-07-30
>
> We thank the Reviewer uJP2's constructive and insightful comments. To address your questions, we provide pointwise responses below.
>
> >* Q1：The proposed method with curve optimization seems to add computational overhead.
>
>   A1: We understand the reviewer’s concern and acknowledge that our method introduces additional computational overhead due to its multi-stage training paradigm. However, we would like to emphasize that this overhead is affordable and justified by the significant benefits it brings.
>
>   First, our approach offers a novel research perspective for MTL, departing from conventional gradient manipulation techniques whose effectiveness has recently been questioned by emerging literature [1,2]. Second, the proposed paradigm enables explicit exploration of task trade-offs, which leads to significant improved performance. Finally, our method is compatible with mainstream MTL approaches, as demonstrated by our plug-and-play experiments, underscoring its flexibility and broad applicability.
>
>
>
> >* Q2: Some of the motivation for the methodology is not clearly discussed. It is unclear why the objective in Section 5.1 is designed in this way. More explanation should be given.
>
>   A2: This objective is designed to imbue the learned curve with trade-off properties similar to those of the Pareto front. During training, by jointly optimizing both the overall task loss $L_T$ and the regularization term $R_o$, we not only ensure a low aggregated loss across tasks but also promote traversal along different trade-offs, thereby facilitating the discovery of optimal task balances. Specifically, we achieve this by encouraging $L_1$ to monotonically increase and $L_2$ to monotonically decrease as $t$ varies from 0 to 1, while maintaining a low overall task loss $L_T$ along the curve.
>
> >* Q3: Mode connectivity and curve optimization is not new. The paper seems to combine and apply existing methods to multi-task learning. It would be better to state the difference of these two in multi-task compared to single-task learning.
>
>  A3: Thank you for your valuable question. We acknowledge that mode connectivity and curve optimization are not novel concepts, and we do not claim them as our contributions. Instead, we provide an overview of related work on these two topics and explicitly clarify the connections and differences between our approach and prior studies in Section 2.
>
>   While mode connectivity has been applied in manifold-based PFL to approach Pareto front, it still faces several challenges. For example, it suffers from degeneration in two-task scenarios and does not scale well to settings involving many tasks. These limitations make manifold-based PFL—what we refer to as **endpoint-based mode connectivity**—less practical for MTL. Therefore, we adopt a **curve-based mode connectivity** approach that directly optimizes the entire curve rather than just the endpoints.
>
>   However, applying curve-based mode connectivity to MTL is not straightforward due to three major issues discussed in Section 4.2: randomness, lack of flexibility, and poor scalability. To address these, we propose three key innovations: an order-aware objective, the adoption of NURBS to define the curve, and a task grouping strategy. Together, these components enable practical and scalable trade-off exploration, leading to improved MTL performance.
>
>   In summary, our main contribution lies in identifying the limitations of applying traditional mode connectivity to MTL and in developing tailored solutions—including new objectives, curve formulations, and task grouping methods—that successfully adapt this framework to the MTL setting and yield strong empirical results.
>
>
>
> >* Q4: Some references should be compared or discussed.
>
>   A4: Thank you for your valuable question. Similar to the principles underlying CAGrad and FAMO, *Steepest Descent* [3] aims to optimize for the worst-case task, thereby aligning with the notion of adversarial task alignment. *MoDo* [4], on the other hand, can be viewed as a variance-reduced variant of MGDA. Both approaches fall within the category of gradient-based MTL  methods and are conceptually orthogonal to our framework. This distinction is further supported by the plug-and-play verification results provided in Appendix B.4.
>
>   We have additionally conducted comparisons with the referenced methods [3, 4] on the MultiMNIST benchmark, reporting task-wise performance under different preferences. As shown, although Pareto MTL and EPO exhibit broad trade-off capabilities, they often struggle to achieve favorable trade-offs, leading to suboptimal overall MTL performance—similar to COSMOS, as illustrated in Figure 6. These comparative results and analyses will be included in the next version of the paper.
>
>   | **EXTRA**          | 0.912, 0.960 | 0.937, 0.957 | 0.940, 0.952 | 0.958,0.949  | 0.963, 0.925 |
>   | -------------- | ------------ | ------------ | ------------ | ------------ | ------------ |
>   | EPO            | 0.640, 0.950 | 0.861, 0.948 | 0.902, 0.880 | 0.933, 0.850 | 0.944, 0.550 |
>   | Pareto MTL | 0.864, 0.900 | 0.892, 0.887 | 0.907, 0.882 | 0.909, 0.862 | 0.918, 0.848 |
>
>
>
> >* Q5: How well would this approach generalize to other tasks/objectives, model architectures with different loss landscape properties?
>
>   A5: Thank you for your valuable question. Mode connectivity is a well-established property, as explored in prior works [5,6,7], and has been investigated across various tasks and architectures. In our evaluation, we further verify its applicability across diverse scenarios, including image classification and scene understanding tasks, as well as architectures such as CNN and MTAN. Intuitively, whenever two or more objectives exhibit differing optimization behaviors and trade-offs are desired between them, our proposed approach remains applicable and effective.
>
>
>
> >* Q6: Limitation: Limited theoretical guarantees on Pareto optimality or convergence.
>
>   A6: We acknowledge that this paper does not provide a formal theoretical analysis of our approach. This is primarily due to the complex loss landscape and the fundamentally different learning paradigm compared to gradient manipulation-based methods. Currently, there is no existing research that offers theoretical guarantees for the convergence of mode connectivity.
>
>   Nonetheless, we would like to offer some intuition in this regard. Consider a two-task scenario as an example. At $t = 0$, we have $\nabla \mathcal{L}_1(\mathcal{C}(0;\theta)) = 0$, and at $t = 1$, $\nabla \mathcal{L}_2(\mathcal{C}(1;\theta)) = 0$. This property arises from using pre-trained single-task weights as the endpoints. Suppose we reasonably assume that these two endpoints are the closest local minima to each other, which is supported by empirical evidence suggesting that mode connectivity works well when endpoints are nearby [8]. This also aligns with the intuition in MTL that tasks should be correlated. If this assumption holds, then it follows that $\nabla \mathcal{L}_2(\mathcal{C}(0;\theta)) \cdot \nabla \mathcal{L}_1(\mathcal{C}(1;\theta)) < 0$.
>
>   Now define $F(t) = \nabla \mathcal{L}_1(\mathcal{C}(t;\theta)) + \nabla \mathcal{L}_2(\mathcal{C}(t;\theta))$. Due to the $C^1$ continuity of NURBS and the differentiability of $\mathcal{L}_1$ and $\mathcal{L}_2$, $F(t)$ is continuous on the interval $[0, 1]$. By the Bolzano–Cauchy Theorem, there exists $t' \in (0,1)$ such that
>   $
>   F(t') = \nabla \mathcal{L}_1(\mathcal{C}(t';\theta)) + \nabla \mathcal{L}_2(\mathcal{C}(t';\theta)) = 0,
>   $
>   which implies the existence of a Pareto stationary point.
>
>   We emphasize that this is not a rigorous theoretical analysis, but rather a conceptual insight. Nevertheless, our contribution lies in introducing a novel perspective to MTL, distinct from gradient manipulation-based approaches. This is particularly meaningful given the ongoing debate over the practical effectiveness of gradient-based MTL methods [1,2]. Furthermore, we properly adapt and apply mode connectivity to MTL, differing significantly from its prior use in PFL.
>
>
>
>   **Reference**:
>
>   [1] Robust Analysis of Multi-Task Learning Efficiency: New Benchmarks on Light-Weighed Backbones and Effective Measurement of Multi-Task Learning Challenges by Feature Disentanglement, arXiv 2024;
>
>   [2] Can Optimization Trajectories Explain Multi-Task Transfer?, arXiv 2024;
>
>   [3] Steepest descent methods for multicriteria optimization. Mathematical methods of operations research, 2000.
>
>   [4] Three-Way Trade-Off in Multi-Objective Learning: Optimization, Generalization and Conflict-Avoidance. NeurIPS 2023.
>
>   [5] Loss surfaces, mode connectivity, and fast ensembling of dnns. NeurIPS 2018.
>
>   [6] Mechanistic mode connectivity. ICML 2023.
>
>   [7] Optimizing mode connectivity for class incremental learning. ICML 2023.
>
>   [8] Optimizing Mode Connectivity via Neuron Alignment. NeurIPS 2020.

---

### Comment · Area_Chair_SXYM · 2025-08-03
**Author-reviewer discussion period in progress**

Dear Reviewers,

Thank you for your efforts in reviewing this paper.

We are now in the author-reviewer discussion period. Given the detailed author responses, we encourage active discussion during this period. If you have not already, please read their response, acknowledge it in your review, and update your assessment as soon as possible.

If you have further questions or concerns, post them promptly so authors can respond within the discussion period.

Best regards,
AC

---

### Decision · Program_Chairs · 2025-09-17

**Decision:**

Accept (poster)

**Comment:**

This paper initially received mixed scores and ultimately reached the upper borderline, leaning toward acceptance. The discussion period was productive in addressing several concerns raised by the reviewers. This AC also shares the strengths highlighted by the reviewers: (1) the paper is well written and clearly organized; (2) it introduces a novel use of curve-based mode connectivity for MTL, which identifies challenges of curve-based mode connectivity and addresses them; and (3) it provides insightful takeaways supported by evaluations and analysis. While some reviewers noted weaknesses, this AC believes the paper’s strengths far outweigh these issues, which can be resolved in a revised version. Accordingly, this AC concurs with the reviewers’ positive views and recommends acceptance of the paper. As a minor note, this AC also observed small typos while going through the manuscript: line 143 (“.” should have spacing before it), line 153 (“staeg”), and several spacing inconsistencies before citations.